# Embolization in Pediatric Patients: A Comprehensive Review of Indications, Procedures, and Clinical Outcomes

**DOI:** 10.3390/jcm11226626

**Published:** 2022-11-08

**Authors:** Paolo Marra, Barbaro Di Fazio, Ludovico Dulcetta, Francesco Saverio Carbone, Riccardo Muglia, Pietro Andrea Bonaffini, Clarissa Valle, Fabio Corvino, Francesco Giurazza, Giuseppe Muscogiuri, Massimo Venturini, Sandro Sironi

**Affiliations:** 1Department of Radiology, ASST Papa Giovanni XXIII Hospital, 24127 Bergamo, Italy; 2School of Medicine and Surgery, University of Milano-Bicocca, 20126 Milan, Italy; 3Department of Vascular and Interventional Radiology, Cardarelli Hospital, 80131 Naples, Italy; 4Department of Radiology, IRCCS Istituto Auxologico Italiano, San Luca Hospital, 20149 Milan, Italy; 5Diagnostic and Interventional Radiology Department, Circolo Hospital, ASST Sette Laghi, Insubria University, 21100 Varese, Italy

**Keywords:** pediatric, endovascular, bleeding, embolization, mini-invasive approach

## Abstract

Embolization in pediatric patients encompasses a large spectrum of indications, ranging from the elective treatment of congenital diseases of the cardiovascular system to the urgent management of acute hemorrhagic conditions. In particular, the endovascular treatment of central and peripheral vascular malformations and hypervascular tumors represents a wide chapter for both congenital and acquired situations. Thanks to the progressive availability of low-profile endovascular devices and new embolic materials, the mini-invasive approach has gradually overtaken surgery. In this review, the main embolization procedures will be illustrated and discussed, with a focus on clinical indications and expected outcomes. The most recent mini-invasive techniques will be described, with hints on the cutting-edge devices and embolic materials.

## 1. Background

Pediatric interventional radiology started to gain acceptance during the 1980s, when the number of practitioners that approached diagnostic and interventional procedures significantly increased [1]. Similar to other radiological procedures, most of the embolization techniques were first developed and widely adopted in adults, before being translated to pediatric patients [2]. If, on the one hand, embolization techniques mirror those performed in adults with the same materials and devices, on the other, pediatric patients pose several challenges, not only due to size, but also to required peri-procedural care and life expectancy. Moreover, attention must be paid to radiation protection. Given the relative rarity of diseases requiring embolization in children, clinical practice guidelines and recommendations are not always available and are often adapted from adult algorithms. 

The aim of this review is to present and discuss the indications, techniques, and clinical outcomes of the main embolization procedures performed in a pediatric referral center, with a set of representative images collected over ten years of experience.

## 2. Methods

A comprehensive review of the literature was performed to include all the relevant papers dealing with the multiple pathologies that potentially involve embolization in pediatric patients, most of which were evaluable in our reference center for pediatric diseases. Lack of high levels of evidence for the treatment of multiple conditions obliged us to range from evaluating international guidelines to retrospective and cohort studies, as well as case series, case reports, and letters to review specific technical aspects and new devices. Classifications of multiple congenital conditions were provided with available references throughout this review article.

Articles were screened in PubMed by titles and abstracts, using the keywords “pediatric ” and “embolization”. Few references of procedures in adults were selectively included when adequate pediatric literature was not found.

## 3. General Considerations Regarding Periprocedural Care

The management of pediatric patients is a team effort, and dedicated and specifically trained pediatric and neonatal intensive care physicians and nurses are vital in peri-procedural care.

The duration and relative invasiveness of interventional radiological procedures in the pediatric population and their potential complications suggest the need for general anesthesia with close monitoring and breathing control. Patient immobility is crucial for safety, radiation dose, and contrast medium saving. As a complication of general anesthesia to be mentioned, severe hemoptysis may occur during the procedures related to positive pressure ventilation [3]. Bland sedation should be reserved to children undergoing easier procedures, such as varicocele embolization or in case consciousness is required for neurologic signs monitoring. The placement of a patient on an angiography table needs accurate evaluation to avoid pressure points and ensure optimal imaging. In neonates, a warming blanket is used to ensure normothermia, and a possible supplement consists of a beanie or warm towel around the head as they lose heat quickly through their scalp. Central venous lines and urinary catheters are mandatory prior to obtaining arterial access or during complex procedures, to aid procedural monitoring and manage eventual complications. Intravenous heparin administration (50–100 U/kg) can be considered only in absence of coagulopathy or bleeding. Post-procedural prophylactic anticoagulation should be applied after vascular plugs implantation (i.e., septal defects, porto-systemic shunts), to reduce the risk of undesired thrombosis [4]. Contrast stagnation in the catheter should be considered, as 1 mL of contrast flushed into a 3-kg baby is significant. In addition to femoral artery injury and non-target embolization, contrast-related risks must be contemplated as procedural hazards. The tolerated dose of contrast emerged to be quietly high in pediatrics, according to previous publications, up to 10 mL/kg of non-ionic medium in life-threatening circumstances, although a lower threshold is recommended (4 mL/kg). Catheter position should be constantly guided by fluoroscopy, with movements that are small and gentle, as for seeing a non-targeted vessel selection, particularly the visceral branches of the abdominal aorta, thus causing spasm, dissection, or perforation. Moving from a guiding catheter to microcatheter may also minimize a vessel wall reaction or injury, as well as intra-arterial glyceryl trinitrate administration [5]. Patients’ foot capillary refill and temperature checking should be evaluated before removing arterial access from the femoral, as the former is as sensitive an ischemia assessment as pedal pulses. Otherwise, operators should gradually remove the sheath or perform an angiogram before removal. External iliac artery spasm is frequently visualized, with internal iliac collaterals providing flow to the lower limbs. The best management consists of removing the sheath and looking for manual hemostasis by soft manual compression, avoiding hemostatic devices. Only when the comfortable hemostasis has reached out, the patient can emerge from anesthesia. After sheath removal, the arterial access site is monitored for bleeding and lower limb capillary refill, as recorded in the ward. When an infant or baby is expected to have ongoing instability, or when the embolization was made under emergency, it may be prudent to leave the baby ventilated another 24 h. It is also quite common for an infant to demonstrate hemodynamic instability after a successful embolization procedure, particularly hypertension. Generally, this is treated aggressively to reduce the risk of cerebral hemorrhage. It is important to mention that, if corticosteroids are contemporary used for post-embolization syndrome, these may worsen hypertension [5].

## 4. Congenital and Acquired Diseases of the Cardiovascular System

### 4.1. Indications

Transcatheter interventional therapy for congenital heart disease (CHD) has made tremendous strides in the past decades [6]. The treatment of congenital cardiopathies is indicated in the first years of life in the presence of heart failure, or later when there are hemodynamic effects characterized by left ventricular remodeling. Timely closure interventions of congenital defects prevent the development of complications, such as heart failure, pulmonary hypertension, arrhythmias, and possibly infectious endarteritis [7]. These clinical conditions can arise from one to multiple coexisting defects. Most single defects include atrial septum defect (ASD) and patent ductus arteriosus (PDA). 

Although surgical treatment has been considered the first therapeutic choice, transcatheter interventional procedures are much less invasive, less expensive, and less painful [6]. Ventricular septal defect (VSD) is the most common cause of CHD, accounting for about 20% of all causes; in particular, the perimembranous (pmVSD) defect accounts for about 70–80% of cases.

Neither randomized clinical trials nor meta-analyses support interventional therapy in the pediatric population yet, so that low grade levels of evidence can be established, even though these procedures have been widely adopted. (level of evidence: 4).

### 4.2. Techniques

General anesthesia is preferably administered to patients aged less than 10 years old, and local anesthesia is opted for older patients [6]. In case of pulmonary hypertension, right ventricular catheterization is performed to monitor pulmonary circulation, total pulmonary resistance, the degree of shunt, and the pulmonary to systemic flow ratio (Qp/Qs). Preoperative antibiotic prophylaxis is administered to prevent infections. Anticoagulation is crucial. A proposed protocol of anticoagulation is based on a loading dose of heparin (100 U/kg) administered intravenously immediately prior to the procedure, as well as one–quarter to one–third of the loading dose hourly during the procedure. Low molecular weight heparin (0.01 mL/kg) is then administered subcutaneously every 12 h for the first 48 h after procedure [6].

Classic simultaneous procedures provide PDA-ASD occlusion. PDA is usually occluded first, before than other coexisting defects. Patients receive arterial and venous punctures, followed by lateral angiography of the descending aortic arch, to observe the morphology of PDA. The diameter and length of the aorta and pulmonary artery are measured. An end-hole type catheter is guided from the right femoral vein to the pulmonary artery. After measurement of the pulmonary artery and right ventricle pressure, the pathway is established by passing a 6F catheter from the pulmonary artery to the descending aorta through the PDA. The delivering sheath is then introduced, and PDA occlusion is performed with the appropriate device. Monitoring aortic and pulmonary artery pressure during the procedure is mandatory. Repeated angiography before device detachment has the aim to observe the morphology of the malformation, as well as the position of the device and any residual shunt. The occluder is released when appropriately positioned with no residual shunts. Afterwards, ASD occlusion may be initiated. A long, stiff guidewire is inserted through the IVC, right atrium, ASD, left atrium, and into the left upper pulmonary vein, followed by a sheath. An ASD occluder attached to the delivery cable is pushed to the top of the sheath and opened after its correct location is confirmed by echocardiography. Echocardiography is useful to observe the ASD occluder position, presence of any residual shunts, and impact of the occluder on the surrounding structures [6]. The optimal size of the ASD occluder is 2–6 mm larger than the maximum diameter of the ASD, as measured by echocardiography. For PDA, a mushroom occluder is commonly selected, with the diameter of the device being 3–6 mm larger than the narrowest diameter of PDA, as measured during aortic angiography [6].

### 4.3. Clinical Outcomes

The criteria for procedural success are given by the occlusion of septal defects (SDs) and PDA, without any emerging adverse events, such as, definite residual shunt, displacement or fall-off of the occluder, or any cardiovascular complications after the procedure [8]. Outcome assessments rely on the ultrasound evaluation of blood pressure and blood flow, which usually tend to restore if the pre-procedural planning is adequate [6]. 

Transcatheter closure of PDA in premature infants is a feasible, safe, and effective alternative to surgical ligation, as well as in extremely low birth weight (ELBW) infants. In order to propose guidelines on management options and potential complications, a single arm, prospective, multicenter, non-randomized study was conducted in the United States to evaluate an Amplatzer plug (Amplatzer Piccolo Occluder; Abbott Structural Heart, Plymouth, MN, USA) to treat PDA in patients ≥ 700 g that yielded an implant success rate of 95.5% overall, with 99% in patients ≤ 2 kg. A proposed modified implant technique also avoided arterial access and exclusively relied on a transvenous antegrade approach guided by fluoroscopy, venous angiography, and transthoracic echocardiography (TTE), with placement of the entire device within an intraductal position to avoid protrusion in the adjacent vessels [9].

These data could more widely support the availability of interventional treatment in pediatric population as a more effective alternative to classical surgical ligation.

However, some speculations may emerge about ASD treatment, considering the results of the systematic review and meta-analysis included in the 2018 AHA/ACC guideline for the management of adults with congenital heart disease [10]. ASD closure had a protective effect on functional capacity, as demonstrated by a significantly higher chance of being New York Heart Association (NYHA) class I after repair, whether with intervention or surgery. Significant improvements in right ventricle systolic pressure, volumes, and dimensions were documented after repair. Finally, improvements in left ventricle function and size in diastole were observed after repair. Although the overall body of evidence was limited to observational cohort studies in adults, that constitutes a low level of evidence because the study limitations, early diagnosis, and preemptive treatment of these conditions may strengthen the evidence on pediatric procedures [10].

Transcatheter occlusion of pmVSD in pediatric patients has been performed more frequently, since the development of the asymmetric VSD packing device, although several diverging evaluations emerged over the years. The first attempts showed high rates of complete atrioventricular blocking during and after the procedure. Besides electrophysiological reason, morphological aspects hindered percutaneous occlusion: a high position of the defect very close to the aortic valve generated iatrogenic aortic regurgitation after disc positioning, due to the right coronary cusp bulging determined by the device. On the other hand, smaller occlusion devices led to a residual shunt. More recently, follow-up data of modified occluders showed long-term alleviated aortic regurgitation, suggesting that early treatment by means of an interventional radiological (IR) approach can be considered effective and safe in these conditions [11].

The sequence of simultaneous interventional therapy for compound defects should follow the principles of “complex first and simple second” and “avoiding adverse effects of the second interventional therapy on the first therapy”, as reported in a small cohort study [6].

Children with CHD often develop abnormal blood vessels that may require transcatheter embolization, the prime examples of which are aorto-pulmonary collaterals (APCs) and veno-venous collaterals (VVCs), mostly found in children following single-ventricle palliative surgery. Collectively, abnormal blood vessels are not only sequelae of surgery for CHD, but may be also congenital themselves, such as, for example, coronary artery fistulae (CAFs) [12].

## 5. Pulmonary Vascular Malformations

### 5.1. Indications

Pulmonary arteriovenous malformations (PAVMs) or fistulas were firstly described by Churton et al. [13], defined as a direct and aberrant vascular connection, called “nidus”, between pulmonary arteries and veins, that bypass lung capillary bed, causing right-to-left shunts. They are rare conditions that require prompt diagnosis and correct management when clinically suspected, as several serious complications may present as consequences of distal thrombosis and embolization by paradoxical mechanism, or malformations rupture, i.e., cerebrovascular stroke or transient ischemic attack, abscesses, splanchnic vessels embolization, hemoptysis, and hemothorax [14]. Most PAVMs are congenital (80%), and a tight association with hereditary hemorrhagic telangiectasia (HHT) is reported. Acquired PAVMs represent 20% of cases, typically associated with hepatic cirrhosis, infectious diseases, traumas, mitral valve stenotic disease, Fanconi’s syndrome, and metastatic cancer. Some sporadic cases of constrictive pericarditis, lung parenchymal cyst, and chronic thromboembolic disease have been also described. A classification was proposed by White et al. [15] on the basis of their angioarchitecture. They differ in the number of afferent segmental arteries: one segmental afferent artery for a simple type and two or more for complex ones. The PAVM sizes influence the degree of right-to-left shunting, so that patients may present with dyspnea, cyanosis, clubbing, or chest pain. Due to decreased filtration of vasoactive substances into the systemic circulation, migraines are a common neurologic manifestation. The diagnosis of PAVM is simple and can be made with computed tomography angiography (CTA). The current indications for treatment include any solitary or multiple PAVMs with a feeding artery diameter higher than 2–3 mm, evidence of growth, paradoxical emboli, symptomatic hypoxemia, or any of the other aforementioned serious complications [8]. Symptomatic children should always be treated, whereas the treatment of asymptomatic children should be considered on a case-by-case basis. The 2020 “Second Intenetional Guidelines for the Diagnosis and Management of Hereditary Hemorrhagic Telangiectasia” (HHT) [16] recommended embolization of PAVMs with a feeding artery of 3 mm or greater, with the caveat that targeting PAVMs with a feeding artery of 2 mm may be also appropriate in some cases [8]. Although older literature stated that cerebral ischemic events did not occur in patients having PAVMs with a feeding artery diameter below 3 mm, subsequent studies suggested that even small ones can result in complications and benefit from treatment [17].

Some authors recommend clinical evaluation with transthoracic contrast echocardiography (TTCE) for both adults and children [18], with PAVM grading in three different levels, depending on the degree of left ventricular opacification after administration of a contrast agent (minimal, moderate, and extensive opacification) and confirmatory CT performed for positive TTCE. These criteria have some drawbacks in the pediatric population, in terms of the false positivity of TTCE, mostly for low grade shunts, thus exposing children to unnecessary or avoidable radiation [19,20,21]. However, according to several studies [22,23], the standard and conservative screening methods may both be acceptable for PAVMs in children. Standard screening with TTCE every 5 years, followed by non-contrast low-dose CT for grade 2 or higher shunts is safe and effective in detecting PAVMs, but it exposes a larger percentage of pediatric patients to radiation. Conservative screening with physical exams, pulse oximetry, and chest radiographs every 5 years is also safe and effective, with the added benefit of decreasing exposure to CT radiation. Furthermore, with the conservative approach, there is a small chance of missing a treatable AVM, which might not be detected until adulthood. Although the existing recommendations state that, for patients with initial negative TTCE screen, screening should be repeated after puberty, and otherwise every 5–10 years, longitudinal studies suggest that PAVMs can grow during puberty, with a rate of approximately 10% per year, and double in size every 5–6 years [24]; thus, a more frequent screening interval during childhood may be appropriate. Significant controversy regarding the treatment approach still remains for the pediatric population. The prevalence and symptomatology of PAVMs in children are similar in frequency and distribution to adults, and serious complications from PAVMs do occur, particularly in PAVMs ≥ 3 mm [25]. 

Endovascular treatment, that is, selective embolization of the pulmonary artery feeding the nidus of the PAVMs, the nidus, or the draining vein, has become a first-line treatment, with advances in interventional devices, even though high-level evidence in pediatrics is still lacking [26]. (level of evidence: 4).

### 5.2. Techniques 

The procedure is commonly performed via a transfemoral or transjugular venous catheterization of the pulmonary arterial circulation. Digital subtraction angiography with high-frame acquisition rates allow us to detect the malformation and to guide its superselective catheterization. The following devices are commonly used, with specific pros and cons. Coils are a well-established option, promoting luminal thrombosis through both the reduction of vascular flow and intrinsic prothrombotic properties of the coil design. They are relatively easy to use, can be deployed via low-profile microcatheters, and adapt exceptionally well to the shape of the vascular lumen. The diameter of the initial coil to be chosen should not be smaller than the feeding artery diameter, as this increases the risk of paradoxical embolization. Amplatzer vascular plugs (AVPs) are other valuable devices. They consist of a dense expandable nitinol mesh, with or without internal polytetrafluoroethylene layers, that reduces the blood flow in a target vascular lumen promoting thrombosis. It is recommended to oversize vascular plugs by 30–50%, relative to the target vessel diameter at the occlusion site. The vascular occlusion induced by AVPs is not instantaneous, it requires spontaneous coagulation and may take several minutes, particularly in high-flow settings. Moreover, conventional plugs require large bore catheters to be delivered. Another option for PAVM treatment is given by the microvascular plug (MVP), which consists of detachable plug with a nitinol skeleton partially coated with polytetrafluoroethylene (PTFE). Its advantages in embolization include microcatheter-based deployment, resheathability, immediate occlusion (even in the setting of intraprocedural anticoagulation), and less metal artifacts, compared to coils. Furthermore, it is thought that the PTFE coating may help to prevent delayed recanalization [8]. Some studies proposed a vascular plug alone or in combination with coils to be a primary option for PAVM embolization when technically feasible [27,28].

### 5.3. Clinical Outcomes 

The efficacy of PAVMs embolization is defined by some authors as >70% sac size regression on follow-up CTA [29]. AVPs were demonstrated to also be very effective for large PAVMs with a feeding artery diameter of ≥8 mm, with no persistence, migration, or complications in follow-up [30]. In 2004, Mager et al. reported long-term outcomes after the embolization of 349 PAVMs in a cohort of 112 patients [31]. The data indicated that patients are much more likely to require reintervention when all visible PAVMs are not embolized at the time of initial procedure. A study by Pollak et al. reported that nearly 20% of small PAVMs will grow over time and that up to half of these may result in symptomatic events or complications [17]. The current evidence suggests that serious complications, including stroke and brain abscess, may occur in PAVMs with feeding artery diameter less than 3 mm [32]. Overall, embolization of all angiographically visible PAVMs at the time of initial procedure, if technically feasible, may significantly reduce the likelihood of needing reintervention and risk of complication [8]. 

Patients with PAVMs who undergo embolization require follow-up at least one month and one year after embolization [33].

A case series reporting almost 7 years of follow-up in pediatric patients helped to recognize significant improvements in oxygenation after embolization, with a low rate of post-treatment adverse events; anyway, it confirmed the persistence of PAVMs in some cases, emphasizing the need for long-term, post-treatment follow-up, and suggested that PAVM-related complications in children tend to occur only with large PAVMs, not posing those small, untreated ones as significant as in adults [8].

## 6. Traumatic, Spontaneous, and Iatrogenic Arterial Bleeding

### 6.1. Indications

Several conditions may determine arterial bleeding in pediatric patients, with trauma being the most frequent cause [34]. Spontaneous hemorrhage is uncommon and may be due to the rupture of hypervascular tumors or vascular malformations, which are described in dedicated paragraphs. Iatrogenic bleedings follow surgical and percutaneous procedures, such as transplantation, biopsies, and catheter insertions [35]. The risk of iatrogenic hemorrhage may be increased by anticoagulant or antiplatelet therapy or by congenital or acquired coagulopathy. The indication for embolization must be shared among interventional radiologists, surgeons, and anesthetists, who together must establish what is the most effective strategy to stop bleeding in the shortest time [36]. A clinically suspected bleeding must be confirmed by imaging, in order to properly identify the source (arterial versus venous) and to plan the most appropriate strategy. In particular, in an emergency setting, CTA provides the identification of the bleeding vessel, facilitating the embolization procedure and reducing radiation exposure. 

On the contrary, imaging evidence of a mild intrafascial hemorrhage or focal intraparenchymal pseudoaneurysms are not indications of embolization per se, unless they occur with hemodynamic instability or when ongoing transfusions are required [37]. Indeed, many arterial bleedings are self-limiting, without the need for any intervention [34]. 

Contrast-enhanced ultrasound (CEUS) has been increasingly used for the diagnosis of post-traumatic lesions in children, due to the absence of ionizing radiations [38]; however, compared to CTA, CEUS lacks a panoramic view, which is crucial to plan the embolization procedure. In our institution, the main causes of hemorrhage requiring embolization in pediatric patients are splenic (Figure 1) and hepatic trauma and percutaneous procedures, such as liver biopsy (Figure 2) and biliary drainage. Acute gastrointestinal arterial bleeding is uncommon in children, although it can be exceptionally observed, due to gastritis or peptic ulcers, Meckel diverticulum, intussusception [2], or intestinal lymphoproliferative disorders (Figure 3).

Embolization is considered the standard of practice for traumatic, spontaneous, and iatrogenic arterial bleeding, although evidence in the pediatric population is still limited. The consolidated experience in adults strongly supports its application in children, even though high-level evidence in pediatrics is still lacking. (level of evidence: 4).

### 6.2. Techniques

Transarterial embolization may precede or follow laparotomy to achieve complete hemostasis [36].

In the case of bleeding, embolization is almost always performed with an endovascular approach via the common femoral artery catheterization [2]. To achieve superselective target embolization and avoid non-target organ ischemia, microcatheters are used to navigate as close as possible to the bleeding site. Embolic materials encompass a wide range of agents, each one with specific properties, which are briefly discussed in the dedicated paragraph. As a general rule, mechanical occlusive agents, such as coils and plugs, should be used in terminal vessels, where there is no chance for collateral revascularization and rebleeding. Particles and liquid embolics are most suited for organs that have a dense network of arterial anastomoses, provided that they are injected superselectively. Bleeding due to arterial ruptures may be managed with vascular grafts. 

### 6.3. Clinical Outcomes

The clinical outcomes of embolization for acute arterial bleeding in trauma are excellent, with high rates of technical and clinical success and low rates of complications and mortality [39,40]. Few data are available on embolization for spontaneous and iatrogenic bleeding in pediatric patients.

## 7. Congenital and Acquired Vascular Malformations (VMs)

### 7.1. Indications

Most of all pediatric cranial and spinal vascular lesions, presenting at any time during childhood, comprise a broad spectrum of pathological types and carry a wide range of risks, including neonatal high-output cardiac failure, developmental delay, seizures, and inherently in younger patients, a higher cumulative lifetime rate of vascular malformation rupture, compared with adults. 

Several classifications have succeeded over years to better define these uncommon lesions, based on histopathologic findings, rather than their vascular dynamics. As a result of the high failure rate and morbidity associated with surgical resection of peripheral VMs, percutaneous embolization has been suggested as the primary therapeutic modality or as a pre-surgical intervention to reduce bleeding and maximize the chances of a successful resection [41].

Among the acquired vascular malformations, post-traumatic and iatrogenic arteriovenous fistulas are worth mentioning. They most commonly occur within abdominal organs (liver, spleen, kidneys) or bones and are characterized by the self-sustaining mechanism of a direct blood flow from high-pressure arteries to lower-pressure veins. Although acquired arteriovenous shunts can lead to acute clinical manifestations, such as bleeding with hemoperitoneoum, hemobilia, hematuria, and melena, they are frequently incidentally discovered via color doppler ultrasound (CDUS).

Arterio-portal fistulas are relatively common in the population of pediatric liver transplants, due to the number of interventions performed in the regular follow-up (biopsies) or to manage transplant complications (percutaneous cholangiography, transhepatic vascular access) [35]. Treatment by means of the interventional radiological approach is considered a standard of care, even though it is yet without a high level of evidence. (level of evidence: 3).

### 7.2. Techniques

The treatment of pediatric patients, particularly infants and young children, whether by open surgical or percutaneous means, poses unique technical challenges. In each case, the risks of treatment-related morbidity and mortality must be weighed against those associated with the natural history of the lesion [42].

The endovascular transcatheter embolization of high-flow peripheral VMs has become more widely accepted as a first therapeutic option; the direct percutaneous approach is reserved to low-flow venous malformations. These procedures require patience and sometimes more than one session, and if performed with the correct material by expert interventional radiologists, they can positively affect the outcome.

A crucial step in the percutaneous management of peripheral VMs is the choice of the appropriate embolic agent, which should be tailored to the morphology and hemodynamic status of the lesion [41].

The common approach to high-flow VMs relies on transcatheter embolization (Figure 4). Selective and superselective arterial catheterization is performed to navigate as close as possible to the “nidus” of the malformation, and embolization is preferably performed with liquid embolics. The primary goal in embolization is to stop the anomalous shunt at the pre-capillary and capillary levels [41]. Arteriovenous malformations may present a direct shunt with systemic circulation, raising concerns about the possibility of performing the embolization with liquid embolics. In such situations, the combined transarterial and transvenous approaches should be considered, in order to obtain the outflow occlusion of the shunt and allow for a safer arterial embolization (Figure 5). The management of small intrahepatic arterio-portal shunts has been also reported with direct transhepatic puncture and thrombin injection [35].

Regarding low-flow VMs, contrast venography is a helpful tool for anatomic characterization, treatment planning to confirm the patency of a normal deep venous system, and to fully assess the extent of the venous malformation and draining venous channels (Figure 6). It is performed once the lesion is accessed at the time of the procedure: the volume of contrast needed to fill the malformation before egress into draining veins allows the operator to determine the amount of sclerosant that can be safely injected [43]. Venous malformations are typically treated with sclerosing agents. The most common agents described in the literature include ethanol, sodium tetradecyl sulphate foam, Ethibloc, polidocanol, and more recently, bleomycin. Sclerotherapy is generally performed with the aid of ultrasound (US) and fluoroscopic guidance (Figure 6). Cone beam computed tomography in the IR suite support multiplanar imaging to evaluate the anatomical distribution of the administered sclerotherapy agent and compare the distribution to the pre-procedure imaging. Fluoroscopy allows to monitor sclerosant injection, in order to decrease the risk of extravasation or non-target diffusion, thereby lowering the risk of complications. Blood pressure cuff, manual compression, and placement of peripheral lines in the affected limb for heparinized saline solution perfusion to reduce risk of thrombosis are commonly used technical tricks. Adjunctive techniques, such as coil or tissue adhesive embolization or laser ablation, may be helpful in selected cases, such as in children in whom rapid venous outflow is demonstrated or in those individuals with large varices or persistent lateral veins, such as in Klippel–Trenaunay KT syndrome. In addition, anastomoses with the normal deep veins can also be occluded with coils or N-butyl cyanoacrylate (NBCA) before the injection of the sclerosant [43].

In patients with microcystic or relatively solid VMs resistant to sclerotherapy and not amenable to surgery, there is emerging evidence to suggest that percutaneous cryoablation may represent a viable and safe alternative for the debulking of large lesions [44,45].

### 7.3. Clinical Outcomes

Dubois [46,47] and Puig [48,49] separately divided VMs into four types, based on the pattern of venous drainage, the variable response to treatment, and the rates of complications. Particularly, Puig’s prospective study and classification established implications regarding the likelihood of a successful response to sclerotherapy [44], with a good level of evidence, although a lack of randomized controls. Types I and II VMs are smaller and respond well to sclerotherapy with higher control rates and fewer sessions of treatment. Higher rates of recurrence and complications are expected with types III and IV [43]. Because of the hemodynamic properties and ever-changing morphology of VMs, their total occlusion, especially the high-flow types, is very difficult to achieve. 

Simply put, VMs demonstrating minimal outflow (Puig type I and II) are associated with better outcomes, primarily because the sclerosant stays within the target lesion for longer. Relatively rapid venous drainage from Puig type III and IV VMs represents a challenge for IRs, as the systemic escape of the sclerosant both impedes the success of sclerotherapy and exposes the patient to the potential for systemic effects of the sclerosant [44].

Therefore, the interventional radiologist should focus on palliation, rather than eradication. Peripheral VMs are associated with severe outcomes, such as the inability to use the involved extremity, cardiac failure resulting from high cardiac output, risk of posttraumatic hemorrhage, and cosmetic/psychological problems in pediatric patients. Maximizing the patient’s quality of life by minimizing these severe outcomes should be the primary goal of endovascular treatment [41].

The most common complications observed after the sclerotherapy of venous malformations are: skin erythema, blistering, skin breakdown, bleeding, necrosis, and hemoglobinuria. Less frequently, thrombophlebitis, thromboembolism, and cardiovascular complications can occur. Hemoglobinuria occurs secondary to hemolysis and can be conservatively treated with intravenous fluid replacement until urinary clearance. Adequate hydration is advised before the procedure [43]. Ischemic complications following the endovascular embolization of high-flow VMs are rarely observed, but should be considered.

## 8. Hypervascular Tumors

### 8.1. Indications

The clinical indications for percutaneous embolization include bleeding and high-output cardiac failure, as well as massive hemorrhage from tumors, such as neuroblastoma and Wilms tumor, in particular, after either trauma or chemotherapy-induced necrosis. The presentation can be extreme, with abdominal tamponade and shock. Embolization has also been reported for palliative care [5].

Although pediatric interventional oncology (IO) practice is based off the medical experience in adults, there exist inherent differences. First of all, cancers in children and younger adolescents are classified by histology into 12 major groups, according to the International Classification of Childhood Cancers (ICCC), in opposition to adult classification, which is based on the anatomical site of the primary tumor. In children, embryonal cancers, retinoblastoma, neuroblastoma, and hepatoblastoma are more prevalent. Additionally, extracranial malignant germ cell tumors (GCTs) are frequent, with a bi-modal occurrence, in infants and adolescents. Other common malignant solid tumors in adolescents are bone and soft tissue sarcomas, melanoma, and thyroid cancer. Anyway, cancers and benign hypervascular tumors, such as non-involuting congenital hemangioma and kaposiform hemangioendothelioma (KHE), are relevant entities [5]. The International Society for the Study of Vascular Anomalies (ISSVA) divides vascular tumors into benign, locally aggressive/borderline, and aggressive tumors [45].

### 8.2. Techniques

Catheter-directed chemotherapy regimens are relatively new in the pediatric setting [50]. Due to developing physiological processes, adult chemotherapy dosing overestimates treatments in younger patients, so that body weight may more accurately predict drug exposure, rather than body surface area (BSA). Small vessel calipers and their early narrowing hinders distal selective catheterization and imposes the use of low-profile introducer sheaths, catheters, and microcatheters. Arterial spasm and dissection are also more common in children, as well as post-embolization syndrome, which often needs good supportive care and longer hospitalization. Despite these impediments, there is an evident increase in the interest for these techniques, as emerging from indexed publications on interventional pediatric oncology [50]. Herein, we provide an overview of the most common hypervascular benign and malignant tumors in pediatric patients.

### 8.3. Benign Tumors

#### 8.3.1. Hemangioma

Hemangioma in an infant is a proliferative cellular neoplasm. The current nomenclature, suggested by Mullikan–Glowaki [51], distinguishes “hemangioma of infancy” from “congenital hemangioma”, which differ in histology and in clinical behavior (the former’s presents expression of GLUT1) [5]. Although these lesions may appear macroscopically similar, in a neonate, a large solitary hemangioma causing high-output cardiac failure is more likely to be a congenital hemangioma than a hemangioma of infancy. They also differ from the “cavernous hemangioma” of adults, which is more similar to venous malformations. Although most hemangiomas require no active treatment other than observation, they may cause problems because of rapid expansion in a critical location, such as near the orbit or airway; anyway, bleeding or high-output cardiac failure from arteriovenous shunting may occur [5,52]. Hemangioma of infancy is the most common vascular tumor in humans, affecting about 10% of Caucasian infants, but it is found in all ethnic groups. It is featured by a rapid proliferation within about 6 weeks after birth, followed by shrinking. The growth allows arterial flow increase, favoring arteriovenous shunting. 

Embolization of hemangioma of infancy is not a first-line therapy, although it has some indications. In particular, embolization can be performed in case of critical lesions that require, but do not respond to, medical therapy (propranolol or corticosteroids), which are either surgically inaccessible or whose excision would be destructive. Besides the above-mentioned common indications to embolization, specific symptoms and syndromes can be associated with hemangioma of infancy. In particular, when lesions are multiple and diffuse in the liver, they may produce a mass effect, not only leading to hepatic failure, but also abdominal compartment syndrome with respiratory and renal failure with life-treating situations [52]. Syndromes such as PHACES (posterior fossa brain malformations, large facial hemangiomas, anatomical anomalies of the cerebral arteries, aortic coarctation and other cardiac anomalies, and eye abnormalities) or PELVIS (perineal hemangioma, external genitalia malformations, lipomyelomeningocele, vesicorenal abnormalities, imperforate anus, and skin tag) require a referral to a pediatric vascular anomalies center for a comprehensive assessment of the disease [5]. 

Hemangiomas fully formed at birth, solitary, and almost 5-cm large correspond to congenital hemangiomas. Their morphology allows us to distinguish them from hemangioma of infancy. Because of a potential extreme flow steal phenomenon, congenital hemangiomas can influence cardiac failure and lead to secondary respiratory renal and hepatic failure. Coagulopathy may also be present, as characterized by thrombocytopenia, high fibrinogen degradation products, and high D-dimer, thus corresponding to reduced fibrinogen blood levels. Kaposiform hemangioendothelioma (KHE) is the other most common vascular tumor, also known as tufted angioma. In contrast with hemangioma, KHE is not GLUT1-positive. It is also associated with thrombocytopenia, due to platelet trapping within the interstices of the mass (Kasabach–Merritt phenomenon—KMP). KHE with KMP warrants treatment, usually chemotherapy, although a role for embolization has emerged. Considering the endovascular approach, the coagulation status of the patient must be carefully taken into account [5].

Almost high-level evidence is available on embolization of hemangiomas (level of evidence: 2). Recent clinical practice guidelines [53] reported numerous studies to have successfully used steroid injection for hemangioma of infancy, such as triamcinolone either alone or in conjunction with betamethasone, demonstrating it to be safe and effective, mostly for those hemangiomas that are relatively small and well-localized, and proliferation may result in increased bulk and threatening anatomic landmarks (i.e., the lip or nose). Larger or more extensive lesions are poorer candidates for this treatment modality, due to the larger volume of steroids required, possible inherent systemic risks, the difficulty of obtaining adequate drug distribution, and the potential for local complications. The guidelines focused on the treatment of infancy hemangiomas with timolol maleate, yet known since 2010 for being significantly more potent than propranolol as a topical application avoiding first-pass liver metabolism, as would occur with an oral treatment. Unfortunately, even in this case, many authors have advocated using limited amounts of medication or have cautioned against application to ulcerated lesions to address concerns regarding potential percutaneous absorption and toxicity [53].

#### 8.3.2. Paraganglioma

Paragangliomas, for their origin from the neuro-ectoderm, refer to different vascular and non-vascular tumors. Some of them constitute carotid body tumors (CBTs), as in the head and neck region they usually are found at the carotid body, jugular bulb, the vagal and tympanic nerves, and the aortic glomus, respectively. The incidence rate in children is lower than adults, approximately 0.02% of all neoplasms, with significant differences in long-term prognosis. Childhood onset has a distinct genetic background, probably from underlying systemic diseases related to genetic mutations (VHL, NF-1, MAX). Hypoxic conditions may be a stimulus for carotid body hyperplasia. It has been observed that individuals who carry CBT susceptibility mutations and are subjected to chronic hypoxia often develop tumors at an earlier age [54].

Diagnosis relies on the location, clinical symptoms, and imaging findings. Imaging plays a crucial role in the diagnosis of CBTs (CTA, magnetic resonance imaging (MRI), US, CDUS, and digital subtraction angiography (DSA)). CTA has the highest sensitivity, revealing a well-defined soft tissue mass with a homogeneous enhancement located within the carotid sheath; CTA provides an accurate mapping of tumor arterial supply. Fine-needle aspiration biopsy (FNAB) is absolutely contraindicated whenever CBT is suspected, due to the high risk of fatal bleeding.

A surgical classification of these entities, based on the degree of involvement of the carotid arteries, was proposed by Shamblin [55]. Type I CBTs are localized and do not compress or involve the carotid arteries. Type II CBTs are adherent to vessels, or partially surround them. Type III CBTs encase the vessels and are considered the most challenging to resect. Complications of CBTs resection are associated with increased tumor size [56,57]. Several systematic reviews and meta-analyses [58,59,60,61,62,63] investigated the role of embolization before surgery of CBTs in adults. Reported advantages include less challenging resection due to reduction in CBT vascularization and consequent decrease of intra-operative blood loss, duration of operation, and complication rates. Perioperative complications, in terms of vascular injuries, strokes, and TIAs, but also cranial nerve (CN) palsy and length of stay, were similar between patients who had preoperative embolization and those who did not. However, some recent studies reported no advantages of preoperative embolization, in terms of surgical outcomes [58,59]. 

The optimal time interval between embolization and surgical resection is another reason for ongoing controversy. Most groups performed preoperative embolization 24–48 h prior to surgery, retaining it to be sufficient for resolution of post-procedural edema. Prolonged intervals may increase the risk for recruitment of other vessels or the recanalization of particle embolized vessels, which could render the embolization a futile procedure [60].

Few different evaluations are mandatory when focusing on pediatric patients. Blood volume is lower, compared to adults, on whom almost all studies were performed. Autonomic dysregulation and proximity to, or infiltration, of critical neurovascular structures also need accurate evaluations. This firstly reflects on surgical and radiological procedures. During angiography, a balloon test occlusion (BTO) should be performed to establish whether the patient can tolerate a carotid artery sacrifice during surgery or preoperatively [54].

Based on literature evidence, preoperative embolization decreases the postoperative mortality rate of 1–2%, but surgical morbidity (32–44%) and mortality (5–13%) remain high. Morbidity of cranial nerve lesions is especially high (40%). In childhood, the surgical removal of CBTs requires advanced skills, due to the smaller intima–media thickness and consequently higher incidence of fatal intra-operative bleeding, even when preoperative embolization is performed [64].

Little evidence exists on interventional management of these neoplasms in pediatric patients; few articles [65,66] report interventional procedures, such as radiofrequency ablation, cryoablation, or ethanol injection, to be used for treatment of a single metastatic lesion or oligo-metastases [65]. 

Because of the rarity of the tumor, a randomized trial is difficult to set up. Given the controversy in the available literature, which is mainly derived from adult studies, the evidence to promote preoperative embolization of paragangliomas in children is still low, and the procedure should be carefully evaluated [63]. (level of evidence: 4).

#### 8.3.3. Renal Angiomyolipoma

Although histologically benign, renal angiomyolipomas (AML) in young people may have more than 50% of risk of rupture and hemorrhage if greater than 4 cm in diameter and rapidly growing, especially when associated with the tuberous sclerosis complex. Transarterial embolization can be considered a safe primary option for symptomatic or large AML, which benefit from tumor shrinkage with reduced risk of hemorrhage; moreover, preoperative transcatheter embolization may reduce the need for blood transfusion during surgery and, sometimes, reduces surgery itself [50].

Radical and partial nephrectomy, transcatheter arterial embolization (TAE), and ablative therapies, including cryoablation and radiofrequency ablation, are considered treatment options among the reported case series [67]. Transarterial embolization (TAE) has been widely performed for AMLs as the first choice for prophylaxis or emergency treatment for bleeding for years [68,69,70]. Several reports have generally suggested that prophylactic treatments, such as TAE, should be performed for asymptomatic AMLs ≥ 4 cm in diameter or AMLs with microaneurysms ≥ 5 mm in the feeding artery [71,72,73]. Symptomatic tumors, such as those with hemorrhage, should always be treated. 

In patients affected by the tuberous sclerosis complex (TSC), current guidelines [73] dictate the management of angiomyolipomas and other renal lesions, such as hamartomas. Before the 2012 approval of everolimus, surgical resection or intravascular embolization were the standard of care for most hamartomas that needed medical intervention. Consensus guidelines proposed in 2012 recommend mTOR inhibitors as a first-line treatment for asymptomatic angiomyolipoma ≥ 3 cm in diameter, with embolization and partial resection reserved as second-line options [74]. (level of evidence: 2).

### 8.4. Tumors with Malignant Behavior

#### 8.4.1. Hepatic Neoplasms

Hepatoblastoma and hepatocellular carcinomas (HCC) account for 1% of pediatric malignancies, with hepatoblastoma alone covering about two thirds of this percentage. Other liver malignancies in children include sarcomas and rhabdoid tumors. Benign neoplasms, such as adenomas, have the potential for malignant transformation, and HCC in children usually occurs in normal livers without underlying cirrhosis. 

The guidelines for staging and treatment decisions for pediatric liver tumors come from three principal international cooperative groups: Children’s Oncology Group (COG), Société Internationale d’Oncologie Pédiatrique Epithelial Liver group (SIOPEL), and the Japanese Pediatric Liver Tumor Study Group (JPLT) [75]. Embolization is a recognized therapeutic option in adults, but its role in pediatric liver tumors has not been validated yet by high-level evidence (level of evidence: 4).

Bland embolization has been used to treat hepatic adenomas, ruptured hepatocellular carcinomas, hepatoblastomas (as pre-surgical treatment), focal nodular hyperplasia, and vascular tumors. A variety of embolic agents can be used, although particles are preferred. The favored size range is 100–300 μm because smaller particles can cause biliary necrosis and bilomas, and larger particles do not cause sufficient tumor ischemia. 

Transarterial chemoembolization (TACE) increases the intratumoral concentration of the chemotherapeutic medication, increasing the dwell time of the medication inside the target tissue and reducing the amount of drug spreading to the systemic circulation. Thus, hypoxia and ischemia induced by TACE further augment the chemotherapy dose achieved within the neoplastic tissue. Traditionally, it can be performed as an emulsion of a chemotherapeutic with an oil-based contrast medium, such as ethiodized oil, but recently, the use of drug-eluting beads has been widely adopted. Indications include neoadjuvant therapy, treatment of non-resectable tumors, and bridge therapy for liver transplantation [76]. 

TACE has a potential role in the treatment of PRETEXT (pediatric liver tumor staging) I and II hepatoblastomas, as neoadjuvant or adjuvant treatment both increase necrosis and potentially reduce relapse, alone or in combination with systemic therapy [50].

Transarterial radioembolization (TARE) consists of transcatheter injection of β radiation–emitting radioisotopes directly into the tumor-feeding arteries through synthetic microspheres, permitting targeted brachytherapy without the systemic side effects of radiation. Mean energy penetration thickness ranges between 2.5 mm and 11 mm. Preliminary planning, based on arterial mapping studies, the embolization of non-target collaterals, and the evaluation of pulmonary shunt fraction, is mandatory. Although limited in availability, this application has been used in the pediatric setting. TARE should be considered for tumors not sensitive to chemotherapy or when chemotoxicity thresholds have been reached. Like TACE, the potential indications of radioembolization in children may include non-resectable liver malignancies or metastases, either as palliation or as a bridge to liver transplantation [77].

The outcome evaluation of embolization firstly relies on tumor volume reduction assessment (according to mRECIST criteria) [78]. Necrosis serves as predictor for patient survival, mostly in hepatoblastomas with undifferentiated small cells [50], for their potential aggressive behavior. 

Regarding major liver surgery, a role for preoperative portal vein embolization to gain hypertrophy of the future liver remnant when inadequate volume is estimated before hepatectomy has been suggested. So far, the literature reported some positive experiences in toddlers with hepatoblastoma and mesenteric hamartoma, with the aim of liver resection, and a recent meta-analysis confirmed the potential role of the technique [50,79].

#### 8.4.2. Wilms Tumor (WT) and Neuroblastoma

Wilms tumor is the most common malignant renal tumor of childhood and 6% of all malignant tumors in children. A diameter larger than 10 cm theoretically makes it unresectable, similarly to the presence of adjacent vital structures infiltration and intracaval/atrial extension [50].

The two major guidelines from the National Wilms Tumor Study Group (NWTSG)/Children’s Oncology Group (COG) [80] and the International Society of Paediatric Oncology (SIOP) [81] do not include embolization among the available treatments, although it emerged to be useful in case series of patients with high-stage disease, those with huge tumors, who had liquefied necrotic area, and those at risk of intra-operative spill and not responsive to preoperative chemotherapy [82,83].

Particularly, pre-operative TACE, associated with short-term systemic chemotherapy, was shown to be helpful in the treatment of primary unresectable anaplastic Wilms tumor, with high rates of complete tumor resection and relapse-free survival [50]. Tumor necrosis after treatment can reach more than 90% in some cases; the literature reports the 5-year event-free survival to be up to 92.7% and overall survival to be 94.5%. No drug-induced cardiotoxicity, nephrotoxicity, or hepatic dysfunction are reported. Bland embolization has been performed in WT to control hematuria after biopsy or to manage life-threatening hemorrhage prior to nephrectomy [50].

Presurgical embolization in large neuroblastomas can be performed to reduce intra-operative bleeding, to treat bleeding after biopsy, and to shrink the tumor to reduce intra-abdominal pressure and compartmental syndrome, as much as for WT. A tumor diameter larger than 13 cm and MYCN gene amplification were found to be two independent risk factors for high-risk tumor rupture and bleeding [50].

The low levels of evidence about neuroblastoma mostly derive from retrospective studies or case reports, and embolization is not universally recognized as a treatment option, unless in selected cases [84,85,86,87]. (level of evidence: 5).

#### 8.4.3. Bone Tumors and Soft Tissue Sarcomas

Malignant bone tumors account for another 6% of all childhood malignancies, 56% of which are osteosarcomas and 34% of which are Ewing sarcomas, with the peak incidence at 15 years of age. Soft tissue sarcoma (STS) covers a large spectrum of neoplasms with more than 100 different histological subtypes of heterogeneous origins and very different natural outcomes. The approach to metastatic sarcomas may vary a lot, according to the patient’s characteristics, histological subtypes of the primary tumor, extension of the disease, in terms of localization, size and number of metastases, the interval between the primary tumor and first metastatic relapse, the interval between two metastatic events, and patient co-morbidities [88]. 

The Society of Interventional Radiology Foundation and the Society of Interventional Oncology recently collaborated to convene a research consensus panel on current state-of-the-art in adult musculoskeletal (MSK) oncologic interventions, potential research, and development opportunities in interventional management and proposed the most important priorities for the interventional oncologic community, in order to advance the collective understanding of primary (not adjunctive) interventional techniques for oncologic applications and improve clinical cares [89].

Little evidence defines the utility of embolization in the pediatric population for this category of diseases. (level of evidence: 5). 

Further evaluations in the future may emerge, taking the risks related to the age of presentation, body volume, and differences on tumor behavior in children into account. The data mentioned below summarize reports from the reviewed literature.

Bland embolization can be used both for local control and symptom palliation, although surgery remains the standard treatment. Transarterial chemoembolization, in combination with limb salvage surgery, yielded encouraging results [50,90]. Chemoembolization with intra-arterial methotrexate has been applied to treat resistant osteosarcoma [91]. Preoperative embolization has proven to be an effective therapy in giant cell tumors, aneurysmal bone cysts (ABCs), osteoblastomas, chondrosarcomas, and vertebral hemangiomas to reduce intra-operative blood loss and decrease transfusion requirements [50].

Even though they are regarded as benign bone tumors of childhood and early adulthood, ABCs present as enlarging osteolytic lesions with a variable potential to be locally aggressive. The interventional radiological approach has been mostly evaluated for lesions with difficult surgical management and to reduce bleeding during excision. 

For extra-abdominal desmoid tumors, doxorubicin endovascularly administered by means of eluting beads has shown promising results [50]. Additionally, the neoadjuvant intra-arterial infusion of cisplatin, pirarubicin, and vindesine for rhabdomyosarcoma and endodermal sinus tumor has been performed. Bland embolization has been used for myofibroblastic tumors, osteosarcoma, and undifferentiated sarcoma to help in preparation for surgery, to treat life-threatening bleeding, and for palliative care. In very young infants, large (>10 cm) sacrococcygeal teratomas pose the risk of rupture and profuse bleeding before or during resection, and preoperative embolization has been safely performed to mitigate the increased bleeding risk. Embolization without surgery also demonstrated a reduction of tumor volumes, ranging from 54% to 97%, over a follow-up interval of 6–32 months [50].

#### 8.4.4. Hypervascular Brain Tumors

The same evaluations extend to hypervascular pediatric brain tumors, such as choroid plexus papilloma, meningioma, astrocytoma, hemangioblastoma, yolk sac tumors, and skull base tumors, which can be embolized pre-operatively to reduce blood loss [50]. The diagnosis of hypervascular brain tumors relies on MRI findings, with strong enhancement, presence of multiple engorged tumor vessels, or intratumoral vascular lakes. The meticulous evaluation of vascular anatomy of the tumor is mandatory to avoid the inadvertent embolization of normal brain, and a microcatheter is advanced into each of the tumor feeders, as close as possible to the tumor stains, to super selectively inject the embolic material. Immediate follow-up control angiogram needs to be performed to evaluate the post-embolization effects.

MRI angiography is useful to identify the major tumor feeders and to plan embolization, which is carried out only in patients who not only have prominent vascular blush, but also at least one visible major arterial feeder. Before embolization, complete carotid and vertebral angiograms are performed for a detailed depiction of tumor vasculature. Based on angiographic findings, anatomical limitations, such as the lack of major arterial feeders, multiple tiny feeders arising from normal major vessels, or tumor encasing major cerebral arteries, may deny an embolization attempt.

The goal of embolization is to achieve obliteration of tumor stains from each tumor vessel to facilitate surgical tumor removal. However, the interventional procedure does not always afford the complete obliteration of tumor stains, precisely for the association between multiple tiny tumor feeders and the sharing of tumor feeders with blood supply to the normal surrounding brain parenchyma.

The embolization of an extra-axial tumor, performed in the territory of external carotid artery, is generally safer than the embolization of intra-axial tumors, which derive their vascular supply from the internal carotid artery or the vertebro-basilar artery. 

Among extra-axial tumors, meningiomas have effective gain from pre-operative embolization in decreasing blood loss, softening the tumor, and facilitating the surgery. 

Among intra-axial tumors, hemangioblastoma is one of the most common brain tumors for which preoperative embolization is performed, while some concerns exist regarding choroid plexus papillomas [88]. The use of particles, such as microspheres and polyvinyl alcohol, or liquid agents, such as glue and non-adhesive embolics, was variably reported, with increased risk of hemorrhage when embolization of meningiomas was performed with particles [88]. 

A smaller circulating volume is critical for potential post-surgical dangers, as massive bleeding and injury to adjacent brain tissue may be catastrophic. Thus, identifying these hypervascular tumors and adopting pre-operative embolic devascularization are critical to the safety and outcome of the surgery [88]. Reported perioperative blood loss widely ranges in the literature, from 50 to 1600 mL, even after preoperative embolization. The reported successful rate of embolization ranges from 50% up to 80%, due to the difficulty in catheterizing tumor vessels, the choice of embolic agent, and the helpfulness in tumor resection [88].

Evidence from randomized trials or large meta-analyses in pediatric patients still lacks definitive support for the role of interventional oncology in these pathologies. (level of evidence: 5).

## 9. Arterial Aneurysms and Pseudoaneurysms

### 9.1. Indications

Pediatric arterial aneurysms and pseudoaneurysm are extremely uncommon. Aneurysms/pseudoaneurysm classification may rely on the etiology (as originally proposed by Sarkar) [92] or on the anatomic site, dividing visceral artery aneurysms (VAAs) (including renal and splanchnic artery aneurysms) from extremity artery ones (including upper and lower extremity aneurysms) [93]. Infection, trauma, connective tissue diseases, non-infectious arteritis, or congenital vascular malformations are the most common underlying causes [93]. The most common cause of visceral artery pseudoaneurysms (VAPAs) are mycotic infections. A specific pediatric population at high risk of arterial aneurysms and pseudoaneurysms is that affected by congenital or chronic liver disease and portal hypertension undergoing liver transplantation. Portal hypertension with splenomegaly is characterized by an hyperdynamic visceral circulation that leads to the spontaneous development of splenic and mesenteric aneurysms; following liver transplantation, the risk of iatrogenic artery pseudoaneurysms should be considered (Figure 7 and Figure 8) [92,93].

Second-level imaging techniques are mandatory for confirming a clinical suspicion and planning the best management, either surgical or by interventional radiology. The aim of non-operative management is to avoid complications and rupture, among all others [94]. Timely evaluation and treatment are essential because they can present with life-threatening bleeding or airway compromise [95]. VAAs and VAPAs management drastically changes, depending on elective or emergent settings, with the latter being affected by higher morbidity and mortality rates. The decision has to take the natural history of the lesion into account, beyond technical parameters, such as the size, risk of rupture, and relative risk of surgical versus radiological intervention. For splanchnic artery aneurysms, we should recognize that we are not, in reality, well-informed about their natural history. This supports the expectancy treatment for most asymptomatic aneurysms. 

The treatment of large or symptomatic VAAs, with a high risk of rupture, and VAPAs is mandatory [94], and the endovascular approach has become the first-line therapy and is less burdened by morbidity and mortality than surgery, with the best cost-effectiveness and requiring a shorter hospitalization. As for adults, embolization in this clinical scenario has been largely adopted, so far, although most evidence in pediatric patients derives from retrospective case series. (level of evidence: 4).

### 9.2. Techniques

The most practical way to treat these aneurysms is transcatheter embolization with parent vessel sacrifice, whereas, in select cases, vessel-preserving procedures can be performed [95]. Both deconstructive and reconstructive procedures may be evaluated based on clinical presentation and angiographic anatomy. Life-threatening conditions require emergent treatment to arrest further deterioration, so the most rapid technique is preferable. Deconstructive procedures, such as endovascular parent vessel sacrifice or endovascular trapping of the aneurysm, showed to be relatively simple, compared to surgery; anyway, inflammatory changes in the peri-aneurysmal tissues, secondary to infection and ongoing thrombosis, can make the surgical approach more difficult. If there are concerns about possible organ ischemia, the target vessel should be temporarily occluded with an appropriately sized balloon, with the collateral supply checked by contrast injection. Clinical evaluation during balloon occlusion is not reliable, given that the effects of ischemia on abdominal organs are delayed. The angiographic visualization of collaterals and organ revascularization could be supplemented with CDUS. Parent vessels can be occluded with either detachable balloons or coils, based on availability.

Reconstructive procedures with parent vessel sparing include stent-assisted coiling or stent graft (covered stent) deployment across the aneurysm. Stent-assisted coiling may not be suited for the treatment of pseudoaneurysms, due to the risk of recurrence and wall erosion, leading to iatrogenic fistula. Balloon-assisted coil embolization strategy is also employed, in case of wide neck saccular aneurysms. Release requires a microcatheter placed within the aneurysm sac contemporary to a parallel balloon catheter inflation along the parent vessel across the aneurysm, thus avoiding coils migration. Coils deployment and balloon expansion allow for the achievement of high-density coil packing. 

A single approach is possible using recent devices for neuro-interventions, such as double-lumen balloon catheters. Flow-diverter stents have been recently introduced for neuro-interventions, but they can be potentially used also for visceral procedures. The use of covered stents is relatively easier and provides almost instant cutoff of blood flow into the aneurysm sac. Favorable factors include the relatively straight segment of the involved vessel, a good landing zone for the stent graft, and the small size of the neck, which can be easily covered by the stent graft. However, the main challenges in the stent graft treatment are the need for relatively large vascular access and the long-term antiplatelet therapy required to prevent stent thrombosis. For the deployment of stent graft, the aneurysmal segment should be initially crossed with a guidewire, which may be difficult or impossible in complex aneurysms [95]. 

Coronary stent grafts (Figure 7 and Figure 8) are a valid option in the pediatric population, since they can be deployed through low-profile guide catheters and introducer sheaths on very thin micro-guidewires. Percutaneous thrombin injection, although commonly exercised in the peripheral vessels, is not advised in the treatment of head and neck pseudoaneurysms because of high chance of recurrence and risk of non-target embolization to the brain.

### 9.3. Clinical Outcomes

After a deconstructive approach, children usually have good vascular plasticity, and collaterals prevent target organ ischemia. 

In the immediate postoperative period, the aneurysm undergoes thrombosis and inflammation. This process can present with a transient increase in the swelling and mild fever, which are usually controlled with analgesics. For the patients who underwent vessel reconstructive procedures, the dose and duration of antiplatelet medications are still controversial, and different institutions follow their own protocols [84]. There are differences in the long-term outcomes between the various arterial territories involved, with visceral artery aneurysm repairs showing high rates of freedom from reintervention among treated patients, up to 83.3% and 69.4% at 1 and 3 years, respectively [93].

## 10. Bronchial Arteries

### 10.1. Indications

The usual indication for bronchial artery embolization (BAE) in children is an immediate threat to life when other treatments have failed. Various authors have proposed thresholds for intervention, based on the volume of expectorated blood. Massive hemoptysis (>8 mL/kg/d), beyond recurrent non-massive hemoptysis and refractory anemia, with diffuse alveolar hemorrhage nonresponsive to medical therapy, are the recognized indications [3]. Because children tend to swallow their sputum, most cases of pulmonary hemorrhage go unnoticed, unless patients present with substantial bleeding or refractory anemia with respiratory symptoms. 

Cystic fibrosis (CF) accounts for most cases in developed countries, whereas infections (tuberculosis, fungal infections, and bronchiectasis of any cause) are more common elsewhere [3]. The diagnostic algorithm depends, to some extent, on the underlying diagnosis. Conventional radiography may rapidly localize the site of angiographic abnormality in about 80% of patients with CF. CTA is recommended for planning BAE, in particular, to confirm pathologically enlarged bronchial arteries and to identify unusual or aberrant arteries and eventual pseudoaneurysms [3]. The Cardiovascular and Interventional Society of Europe (CIRSE) standards of practice on bronchial artery embolization do include the evaluation of pediatric populations, even though, it is based on a single retrospective study [96], thus confirming the paucity of standardized knowledge [97]. (level of evidence: 4).

### 10.2. Techniques

Aortic angiography is performed through the femoral artery with the use of a 4–5F Pigtail catheter to identify the aortic origin of bronchial arteries. The choice of the angiographic catheter relies on the “rule of 110”, according to which the length and width of the catheter tip should be approximately 110% of the width of the aorta at bronchial artery origin. Selective angiography of orthotopic bronchial arteries is commonly performed with 4–5F Cobra-, Simmons-, or Mikaelsson-shaped catheters. Selective angiography of supra-aortic vessels, in order to look for aberrant supplementary vessels is usually performed by a vertebral- or headhunter-shaped catheter. Super-selection of abnormal arteries requires coaxial microcatheters with a 1.8–2.8F profile. Brachial or axillary access may be required depending on “difficult” anatomy [3]. The following angiographic findings are targeted for embolization: enlarged or tortuous vessels, pulmonary parenchymal hypervascularity, bronchial-to-pulmonary shunting, and bronchial artery aneurysms [98]. The embolic materials include polyvinyl alcohol (PVA) particles ranging in size from 300 to 700 μm, microspheres ranging in size from 100 to 700 μm, and 0.035- or 0.018-inch micro-coils. Some authors suggest that particles should be mixed with a 50:50 contrast agent/saline solution at a ratio of 1:2 [98] and that heparinized saline (1 U/mL) solutions should be preferred to fully anticoagulation [3].

### 10.3. Clinical Outcomes

The failure of conservative treatment to control life-threatening hemoptysis elicits the reoccurrence of BAE, even though the evidence is still weak. BAE is a routine intervention to control massive hemoptysis in children with CF, while reports of BAE in those pediatric patients without CF are exceptional. Even though drug therapy alone can achieve immediate hemostasis, the short-term bleeding recurrence rate can be as high as 30% [3]. BAE disadvantages include its invasive nature and severe complications onset, such as the non-target embolization of spinal vessels that commonly originates from the proximal tract of bronchial arteries. Additionally, the expertise needed to perform BAE is not always available in children’s hospitals [3]. 

Technical success is defined as the ability to catheterize and embolize the abnormal bronchial or non-bronchial arteries that are responsible for the bleeding. The development of more meticulous techniques and superselective embolization has raised the percentage of technical and clinical success up to 90–100%, with the latter being defined as the complete cessation of hemorrhage or significant reduction in hemoptysis after BAE, without requiring further intervention for at least 24 h or within 30 days. Technical failures are usually caused by inability to achieve a secure and stable catheter position in BA, failed embolization in extensive and bilateral disease, or not recognizing the pulmonary artery as an origin of the bleeding [97].

The reported rate of major complications of BAE is low, although it may be biased, as the number of reporting centers is considerably lower than the number where BAE is performed. Several types of neurological complications have been reported, ranging from peripheral nervous branches involvement, as for phrenic palsy (embolization of the pericardiophrenic branch of the internal thoracic artery), to spinal cord ischemia (usually temporary, although small risk of permanent paraplegia is reported) or even fatal brain injury. Various potential mechanisms could cause inadvertent CNS embolization: as mentioned, the passage of embolic agents, either through communications with pulmonary veins or intra-cardiac right-to-left shunts, or aberrant arterial communications with the vertebral circulation [3]. 

Possible collateral damages include myocardial injury (embolization through bronchial artery to coronary artery anastomoses) and finger-tip ischemia (passage of particles through collaterals to the subclavian circulation). The reflux of a small quantity of particles into the aorta may cause sub-diaphragmatic complications, such as bowel ischemia. Ischemic bronchial complications, such as stenosis, infarction, and broncho-esophageal fistula, have been also reported [3].

Immediate control of hemoptysis can be expected in 85% or more of patients with CF. Most of them have more than a year free from hemoptysis after BAE. The relapse rate depends on the definitions, but patients should be warned that repeated BAE procedures are eventually needed in about 30–40% of them. Early mortality is higher than in CF controls without hemoptysis. The success rate for other indications is probably better than 95% [3].

## 11. Lymphatic Embolization

### 11.1. Lymphatic Malformations and Cysts

#### 11.1.1. Indications

As congenital dilated lymphatic channels, lymphatic malformations (LMs) are filled with a proteinaceous fluid and generally do not have connections to the normal lymphatic system. Lesions can be macrocystic, microcystic, or mixed. They appear as soft, non-pulsatile masses with normal overlying skin present at birth or early childhood. Overlying angiokeratomas can further increase bleeding and infection rates. LMs occur in the head and neck in 48%, trunk and extremities 42%, and intra-thoracic or intra-abdominal viscera in 10%. 

Potential compression of adjacent structures, such as airways, vessels, and nerves, may require rapid therapeutic strategies. LMs are generally defined as macrocystic or microcystic, based on cyst sizes of more or less than 2 cm. LMs can be solitary or multifocal, slow growing, and rarely involuting [43]. 

Imaging relies on high contrast resolution modalities and tissue characterization, which confer US and MRI the best chances of diagnosis and management: classic anechoic cavities with internal septa and debris are typical of macrocystic lesions, while small cavities result in innumerable reflective interfaces and a hyperechoic appearance, thus with a more solid appearance of microcystic ones. MRI shows predominantly fluid-type characteristics on all sequences (low signal on T1 and high signal on T2 sequences), with varying degrees of septation and fatty elements. MRI is an excellent modality to assess lesion extents, in terms of tissue planes, airway compression, mediastinal extension, and potential solid organ, and bone involvement [43]. The available literature promotes embolization for these diseases, with an almost high level of evidence [99]. (level of evidence: 3).

#### 11.1.2. Techniques

Sclerotherapy for lymphatic malformations and cysts is generally performed with US guidance, which provides good visualization of the lesion during access and reduces the number of puncture attempts. Contrast is injected under fluoroscopic guidance to depict the entire cystic component and to rule out any abnormal communication with the vascular system. 

Larger macrocystic lesions are generally treated using percutaneous catheterization to allow for the adequate drainage of the cyst and refill with sclerotherapy agent. In patients with large head and neck lesions with the potential for compression, the procedure is generally performed with general anesthesia, and the child remains intubated and monitored during the course of the therapy [43]. 

Microcystic LMs are more difficult to treat, since multiple septations limit the spread of the sclerosing agent to the separated cystic components. Multiple subsequent US-guided punctures may be required, with significant increases of procedural time and required technical skills [43]. 

#### 11.1.3. Clinical Outcomes

In general, macrocystic lesions are more responsive to any sclerotherapy agent than microcystic or mixed lesions, with reported response rates of 88–100% using agents such as ethanol, doxycycline, OK432, and sodium tetradecyl sulphate [43]. Microcystic lesions are a challenge to treat and generally require multiple therapies. Agents such as sodium tetradecyl foam, doxycycline (bland or foam), and bleomycin have been directly injected into these lesions, with reported response rates between 55.7 and 100%. Response rates for OK432 in microcystic diseases have also been reported as 68%, compared with 90–100% for single cysts and macrocystic diseases [43,100,101].

### 11.2. Lymphatic Leakages, Chylothorax and Chylous Ascites

Lymphatic alterations also include leakages that may present as chylothorax or chylous ascites (Figure 9). These uncommon conditions have an uncertain incidence in the pediatric population. Typically, chylothoraces are classified as traumatic or non-traumatic. As uncommon and potentially morbid and life-threatening, they usually arise after cardiothoracic surgery. Congenital syndromes, malignancy, and benign conditions constitute non-traumatic cases. Diagnosis relies on laboratory tests on lymph, hematic triglyceride, and cholesterol values, as well as electrophoresis; so, interventional radiological eligibility: fluid triglyceride > 110 mg/dL, daily leak volume over 250 cm^3^ for 3 days, medical therapy (fluid drainage, dietary modification and octreotide) failure in decreasing the leakage after 10 days are the indications for interventional procedures. Otherwise, contraindications include known pulmonary insufficiency or right-to-left shunts [102]. If the chylothorax is unremitting, despite maximal medical therapy, thoracic duct embolization can be performed with the different described techniques [103,104,105,106]. Despite the literature recognizing lymphangiography and thoracic duct embolization as safe and effective alternatives to surgery in adults, the pediatric thoracic duct embolization and disruption literature is limited to some case reports and a case series [107], without standardized evidence. (level of evidence: 4).

Protein-losing syndrome (PLS) and enteropathy (PLE) can be due to lymphatic obstruction, leading to dilatation, rupture, and opening of the lymphatic vessels into the intestinal lumen. Once PLE is suspected, it is recommended to distinguish whether the disease is of primary or secondary origin. Echocardiography and abdominal imaging are generally required, since the secondary forms are usually cardiac or associated with tumors, such as lymphoma, sarcoma, and neuroblastoma. Central venous pressure is often elevated above the thoracic duct pressure in patients with right-sided heart failure or single-ventricle circulation, leading to obstruction of the lymphatic flow [108,109,110]. Focal short-segment intestinal lymphangiectasia can be treated via intestinal resection or radiologic embolization after dietary therapy failure [110]. The evidence that supports embolization in these conditions is still low. (level of evidence: 5).

#### 11.2.1. Techniques

Direct US-guided retrograde access of the thoracic duct at the venous angle has been described, as has transvenous retrograde catheterization via the brachial vein [111]. Alternatively, bi-inguinal intra-nodal lymphangiography, using a 25- to 30-gauge spinal needles and hand injection of ethiodized oil, can be performed (Figure 9) [112]. Fluoroscopy and spot radiographs are used to record the oil progression from the inguinal lymph nodes through the pelvic lymphatic chain and into the retroperitoneal lymphatic vessels. Treatment continues depending on the visualization of a targetable retroperitoneal lymphatic vessel, cistern chyli, or the thoracic duct, which allows for the switch to direct transabdominal access with greater needles under fluoroscopic guidance. A guidewire is advanced to obtain a central lymphatic access, and the microcatheter is navigated on the guidewire into the lymphatic channels. Digital subtraction lymphangiography is performed to evaluate any lymphatic injury or alternative lymphatic abnormality explaining the chylothorax, and cone-beam CT is used to better evaluate the lymphatic vessels and the site of leakage. Thoracic duct embolization is then preferably performed with liquid embolic agents (NBCA, ethylene vinyl alcohol (EVOH) copolymer) and micro-coils. Simultaneous anterograde and retrograde approaches could increase the technical success of thoracic duct embolization in case of its discontinuation. The pre-procedural antibiotic prophylaxis is weight-based. Post-procedural management relies on chest drainage tube maintenance (until output reduction), non-fat diet, and octreotide until complete clinical resolution [102].

#### 11.2.2. Clinical Outcomes

Clinical success is defined as the complete resolution of chylothorax or any other chylous leak-related condition [102]. 

Although performing these procedures in children is challenging, given the recommended dosages of ethiodized oil (up to 0.25 mL/kg or less more) and the size of lymphatics, relative to available interventional devices, in comparison with adults, the decreased body size may allow for shorter procedure time because the transit of ethiodized oil is faster and the smaller anteroposterior diameter allows for shorter needles with less tissue between the skin and central lymphatics [102]. Some authors reported the clinical efficacy of lymphangiography, embolization, or disruption in about 64% of procedures. Treatments were reported to be well-tolerated by children, without major complications and with acceptable minor complications [102].

## 12. Porto-Systemic Shunts

### 12.1. Indications

Congenital porto-systemic shunts or fistulae are rare conditions and today more frequently identified, thanks to the increased use and technical improvement of CDUS and the development of the CTA and magnetic resonance angiography. The most recent classification recognizes intrahepatic shunts located between the portal vein or one or several of its branches, and the inferior vena cava or a hepatic vein (Figure 10), including the ductus venosus and extrahepatic shunts that directly join the the spleno-mesenteric vessels to the inferior vena cava or one of its branches (e.g., iliac veins) [4]. Clinically significant complications of these shunts are liver tumors, pulmonary arterial hypertension, hepato-pulmonary syndrome, and encephalopathy, caused by the liver bypass of splanchnic venous blood draining directly into systemic circulation. The management in children is still controversial, as some intrahepatic shunts may regress spontaneously, while some others may remain asymptomatic for long periods of time [113]. The congenital extrahepatic portosystemic shunt (CEPS), also known as Abernethy malformation, is the condition with the worst prognosis, depending on the degree of development of the intrahepatic portal circulation. Anatomically, two main types of CEPS have been classified: an end-to-side portocaval shunt with or without an ectopic portal vein; and a side-to-side shunt which is characterized by hypoplastic intrahepatic veins [114]. 

As for other types of hepatic shunts, indications for treatment rely on anatomical and clinical aspects: location and dimensions of the shunt and persistence or growth of the shunt with hemodynamic changes during the follow-up [35]; acute clinical manifestations or clinical signs of severe heart failure or pulmonary hypertension, hepatic encephalopathy, and hepato-pulmonary syndrome [114]. While the majority of such episodes can be managed with usual medical measures, their frequent occurrence resulting in significant disability warrants the consideration of invasive measures, that include surgical ligation or embolization, with the latter preferred, thanks to its reduced invasiveness [115]. 

Evidence is also pushing toward the indication of preemptive shunt embolization in asymptomatic patients, but there are still ethical concerns regarding the invasive treatment of a potentially non-evolving condition: given the rarity of the disease, data are being collected in an international registry (IRCPSS) to investigate all these aspects.

However, the lack of a consensus statement is due to the rarity of the condition, anatomical vascular variants, location, and clinical complication, thus limiting the evidence to some case series and literature review. (level of evidence: 4).

### 12.2. Techniques

The procedures are performed under general anesthesia by dedicated pediatric anesthetists in the angiographic suite; anesthesia is maintained under ECG, blood pressure, pulse oximetry, temperature, and capnometry monitoring. The internal jugular vein is accessed, and a 4F multipurpose catheter is advanced into the portal vein through the fistula. 

After a diagnostic portal venography, a larger sheath is inserted in order to perform the occlusion test, which is recommended to rule out the development of portal hypertension secondary to shunt occlusion and to assess the morphology and the hemodynamics of the intrahepatic portal circulation. If the test is negative, the shunt can be occluded in the same procedure, without expected clinical consequences. If the test confirms the development of high portal pressure during balloon occlusion, the shunt embolization must be performed in two steps: first, a partial occlusion of the shunt is followed by complete embolization several weeks later in a separate procedure. This strategy is aimed to minimize the risk of possible complications, such as portal thrombosis, gastrointestinal bleeding due to varices rupture, and ascites [4]. 

Percutaneous embolization may be challenging in difficult anatomic situations, in particular, in cases of short and wide extrahepatic shunts. Although no specific radiological devices are designed for porto-systemic shunt embolization, plugs and coils have been commonly used in this application (Figure 10) [113]. Some authors also reported the use of custom-made devices, which were modified to allow for the execution of two-step shunt embolization; these include tailor-made reducing stents [116], perforated ventricular septal occluder with imbricated stent graft [117], and cut microvascular plugs imbricated with coils [118].

### 12.3. Clinical Outcomes

The technical and clinical success of embolization was usually obtained in the majority of the reported cases. Procedural risks are related to lack of exact localization of the shunt or anatomical complexity of the shunt. Portal vein thrombosis, gastrointestinal bleeding, ascites, sepsis, and occluder device migration have been occasionally reported [119], but without mortality. Long-term follow-up reports confirm the benefits of minimally invasive radiological techniques in the management of patients with recurrent hepatic encephalopathy who previously responded poorly to medical management [115].

## 13. Sclero-Embolization of Male Varicocele

### 13.1. Indications

The biological approach clearly states that boys with treated varicocele show catch-up testicular growth and improvement in semen quality [120]. According to the literature, since 1995, Kass and coworkers [121] identified some of the indications to treat varicocele: abnormality of semen analysis; left testicular volume at least 3 mL lower than that of the right; secondary sexual hormone axis hyper-function (i.e., LH/FSH response to Gn-RH); presence of bilaterally palpable varicoceles or large symptomatic varicocele. It is not clear whether to treat or observe pre-pubertal varicocele. 

The previous expectant approach recommends observing the pathology over time, unless documented atrophy appears, and eventually treating it in adolescence. 

Noteworthy, if varicocele in adolescence has not to be screened for renal tumors, an infant with varicocele needs abdominal imaging to assess for tumors, classically Wilms and some benign conditions, mainly anatomical, up to situs viscerum inversus [120]. Precisely, thanks to the widespread availability of CDUS (Figure 11), the World Health Organization (WHO) expanded the grading system to include “subclinical” or Grade 0, on a scale from Grade 0 to Grade 4, detectable only by ultrasound, which corresponds to not palpable, as well as during the Valsalva maneuver [122]. The support of ultrasonography relies on the measurement of testicular volume (using the Lambert formula); Doppler measurement of the peak retrograde venous flow (PRF) in the spermatic cord, which emerged as a predictor for progressive testicular asymmetry, if greater than 38 cm/second; measurement of the maximum pampiniform plexus vein diameter (MVD) during Valsalva, as a prognosticator for semen parameters, following varicocele repair, even though estimated only in adults, with an MVD greater than 3 mm preoperatively associated with favorable outcomes [122].

High levels of evidence that favors embolization come from several meta-analyses that analyzed both surgical and interventional treatments. (level of evidence: 1).

### 13.2. Techniques

A plethora of different types of treatment are described in the literature, ranging from laparoscopic and endovascular procedures to open surgery. All of them have the goal of occluding the testicular veins or pampiniform plexus, varying on the anatomical site of occlusion, from distal spermatic cord (low inguinal) to proximal testicular vein (retroperitoneal). Multiple variations in surgical techniques encompass arterial- and lymphatic-sparing approaches to reduce the risk of testicular atrophy and hydrocele, respectively [122]. The main advantages of percutaneous embolization are that it naturally spares the arterial and lymphatic bundles and minimizes the risk of infections. Concerns with the radiological approach include exposition to ionizing radiation and contrast agent administration.

Pre-procedural outpatient appointments allow us to clearly explain the nature of embolization technique, the anatomy of the varicocele, and potential complications or reasons for treatment failure, mostly for patients with previous surgical failure [120]. 

Interventional approaches provide both superior and inferior accesses. The superior one usually corresponds to the right internal jugular vein, right basilic, or brachial vein: this choice offers better angles to enter either the left renal vein or the right spermatic vein. By this way, a standard 4F sheath is generally adequate. The inferior approach may require larger sheaths for coaxial systems, due to the complexity of the angle. Some commercial 6F catheters (i.e., RDC guiding catheter) have a double curve that more easily allow to cannulate left renal vein and the left spermatic vein. Coaxial placement of a standard 45°-angled 4F catheter is then facilitated through the guiding catheter, allowing for the distal cannulation of the vein, without the need of more expensive microcatheters. 

Microcatheter usage is evaluated in some conditions, as venous lumen narrowing or spasm during cannulation. A selective venogram of the renal vein during the execution of the Valsalva maneuver is performed, in order to detect the origin of the spermatic vein. After cannulating the spermatic vein, another venogram during Valsalva is acquired to confirm the presence of varicocele and provide a comprehensive picture of its anatomy. Guidewires are used to reach the spermatic ring and to navigate the catheter up to the more distal aberrant feeding vessel [123]. 

There are many methods to occlude the spermatic veins. The distal spermatic veins at the internal inguinal ring are usually occluded by coils, as for major tributaries if several collateral branches are present. The minor branches are typically treated with sclerosants. High variability in the methodology is found in the literature with practitioners that place coils along the entire length of the vein or others that prefer to use only a sclerosant agent. More recently, the use of liquid embolics, such as cyanoacrylates, has been also proposed (Figure 12 and Figure 13) [124,125]. The external compression of the spermatic vein at the inguinal ligament, manually or by an adequate device, prevents the sclerosant from flowing into the sub-inguinal portion of the varicocele, which could potentially lead to phlebitis. 

The advantage of liquid embolics and scelorsants, compared to coils, is their ability to diffuse over the pathologic vascular network to permanently occlude all the veins by causing profound intimal injury. It is particularly important to evaluate splanchnic or paravertebral variant vein communications, such as an injection of sclerosant, can lead to portal vein thrombosis [124]. Right spermatic vein collaterals recurrence is estimated in 25% of left varicocele. 

The risk of this kind of recanalization can be reduced by empiric sclero-embolization of the right spermatic vein [120]. The recurrent varicocele strictly necessitates focalizing on missed duplicated spermatic veins and collateral pathways. From this point of view, the usage of coils may represent a disadvantage, which may mechanically prevent the cannulation of the vein, should a subsequent procedure be required. Moreover, in pediatric interventional radiology, permanently implanted devices, including coils, should be avoided if there are equally effective nonpermanent agents [120]. Finally, radiation dose reduction techniques are used whenever possible to downgrade the significant risk of over-irradiation.

### 13.3. Clinical Outcomes

The best quality of evidence for the outcomes of varicocele embolization is offered by the latest systematic review and meta-analysis provided by the European Association of Urology/European Society of Pediatric Urology (EAS/ESPU) [126], which included 12 datasets of a total of 16,130 children and adolescents ≤ 21 years of age. Both surgical and interventional treatments were included. The outcomes assessed were a short-term cure or success (evaluated at <9 months), testicular catch-up growth, pain resolution, sperm parameters, and paternity (evaluated at >12 months) for benefits, as well as complications such as testicular atrophy, hydrocele, wound infection, and failure rate for harms. The success rate (disappearance of varicocele) was between 87% and 100% among the randomized control trials (RCTs). The superiority of a specific treatment approach was not identified, probably justifying the less invasive approach [127].

Procedural risks include pampiniform phlebitis, venous thrombo-embolism, failure of procedure, recurrence of varicocele, or infection, as well as extremely small risk regarded by testicular injury or infarction [120,128].

## 14. Technical and Procedural Aspects

There are a wide range of vascular interventional radiology procedures that have either subtle or major variations when performed in the pediatric population, compared with adults. Knowledge of these procedures, the variations, and the complications are essential for a radiologist involved in any center that deals with pediatric patients. Firstly, in the pediatric population, obtaining vascular access becomes progressively more complex as patient age and size decrease [2]. 

### 14.1. Vascular Access, Sheaths, Catheter, and Microcatheters

As a general rule, the size of the vascular access should be as small as possible to achieve the target, at least for arterial embolization interventions. To reduce the risk of vascular complications (dissection, pseudoaneurysms) at the access site, a US-guided puncture should be preferred over freehand techniques. For arterial interventions the common femoral route should be preferred to minimize the risk of thrombosis; US-guided puncture is recommended to avoid inadvertent puncture of the superficial femoral axis. In case of high-flow abdominal shunt, the aorta and femoral vessels distal to the shunt may be hypoplastic, so that brachial access may be required. For venous interventions, the jugular or femoral routes should be chosen on an individual basis. 

Micropuncture sets featured by 0.021” needles and 0.018” guidewires should be used to gain vascular access before insertion of the introducer sheaths. The profile of the introducer sheath varies, according to the coaxial catheter caliper required for the delivery of a specific device, ranging from 3.3F up to 8F. Literature suggests a 4F arterial sheath as largest femoral access, if required, in babies under 4 kg, with 6F for venous ones. Sheath usage should lower intimal trauma from endovascular movements, mostly in those vessels prone to spasm. Another suggested option is fashioning a smaller sheath using the outer sheath of a 4F micropuncture access kit connected to a hemostatic valve to gain analogue 4F catheter with an 0.035-inch inner lumen and smaller outer diameter. Large-sized microcatheters can then be used as primary angiographic catheters, as well as 2.7F or 3F arterial sheaths with a 3F end hole angiographic catheter with the tip steamed into shape. For arterial interventions in the first few days of life, access may be secured through the umbilical artery, keeping in mind that the umbilical artery in a baby over 2.5 kg should accept a 5F sheath. Consensus for consultations among neonatologists, pediatric interventional radiologist, and cardiologist is necessary [5]. Particular attention must be also paid to the thrombosis in the catheter’s dead space and hematic loss during catheter exchange, especially in neonates and babies: small thrombi and air bubbles may produce fatal non-target embolization. To prevent this, the use of continuous saline perfusion through Y connectors and hemostatic valves, as performed in neuroradiologic intervention, is recommended. Considering technical developments that led to the availability of 0.010” coils, 0.021” microcatheter compatible microvascular plugs, and liquid embolics, the catheter profile required to perform transcatheter embolization has significantly dropped in recent years, and most of the devices used in the adult population can be safely adapted for the pediatric use.

### 14.2. Embolic Materials

The choice of embolic materials is a great challenge in pediatrics, mainly regarding liquid embolics, because of the strict relation between available volume and patient weight. Some specific features of the different embolic materials have already been briefly discussed in previous paragraphs. Herein, we present a summarizing overview.

#### 14.2.1. Sclerosants

Sclerosants have been used in vascular and nonvascular settings, both as primary and adjunctive therapy. Beyond technique and delivery, understanding drug properties, in terms of uses, limitations, dosing, and side effects, is required [129].

Sclerosants are, by nature, injectable chemical cauterants intended to scarify and obliterate tissues in a destructive process intrinsically associated with potential complications and outcomes that are not always easy to predict or precisely control. Indeed, most chemicals capable of ‘‘denaturing’’ living tissue are also capable of producing uncontrolled thromboses, unintended destruction of non-targeted vascular tissue, tissue necrosis, anaphylaxis, and neurologic phenomena [130]. The major application of sclerosants is the treatment of cystic and lymphatic malformations; injection can be performed directly into the lesion through the needle or via catheter. Since its introduction as a sclerosant, ethanol has been the standard to which all other sclerosants have been compared. Its mechanism of action is a combination of cytotoxic damage induced by the denaturation and extraction of surface proteins, hypertonic dehydration of cells, and coagulation and thrombosis when blood products are present. All of these factors lead to fibrinoid necrosis. Ethanol’s deep penetration into the vascular wall and lack of viscosity allows it to act in most tissues, without specificity. For this reason, the use of ethanol in neurologically sensitive areas should be avoided [129]. The maximum volume of absolute ethanol that could be safely used in one procedure is 1 mL/kg [5].

Doxycycline is another widely used agent, whose indications assert that the instillation protocol should provide 1–3 treatments for consecutive days, with an instillation time of 1–6 h and a concentration of 10 mg/mL to a maximum of 150 mg in any single session, due to possible major immediate complications for higher doses: hemolytic anemia and metabolic acidosis deriving from hypoglycemia. Doxycycline showed less neurotoxic effects and local skin risks (blistering and skin necrosis), compared to absolute ethanol. 

In 1987, Ogita et al. firstly reported usage of OK432 [101], corresponding to an attenuated strain of Streptococcus pyogenes; then, other authors demonstrated its efficacy on macrocystic lesions with excellent response rates of 88% and low rates of adverse events [100]. 

Bleomycin is a more controversial alternative, as it counts within anti-neoplastic drugs. The biggest concern is pulmonary toxicity, and the strong correlation between toxicity-related risk of death for total doses superior to 450 mg, assuming a dose regimen of about 15 mg/m^2^ twice per week in case of repeated treatments as previously reported [43,44].

With their introduction in the 1920s and 1930s, detergent sclerosants, also known as fatty acids and fatty alcohols, soon became (and still are) the most popular sclerosant types worldwide for their increasingly favorable risk-to-benefit ratios in the endovascular use. Detergent sclerosants produce endothelial damage by multiple mechanisms associated with a decrease in endothelial cell surface tension, interference with cell surface lipids, disruption of intercellular cement, and extraction of cell surface proteins. Detergents are most effective in the form of micelles (molecular aggregates) [130]. The ability to be agitated and foamed increases the power of detergents by up to four-fold, to mechanically displace blood, and to maximize the surface area and time in contact with the endothelium. This process has the advantage in using small, and presumably less, allergenic or less tissue-toxic concentrations and volumes of sclerosant, although greater risks may occur if the foam passes through a patent foramen ovale (incidentally found in approximately 25% of the population) and spreads to the ocular or cerebral circulation. Temporary ischemic attacks associated with migrainoid visual disturbances, amaurosis, and strokes have been reported [130]. A commonly used detergent sclerosant is 3% sodium tetradecyl sulfate (STS), mixed 5:1 with ethiodol and added with 5 to 10 mL of ambient air via a three-way stopcock to generate the foam. STS foam tends to form a cast of the vessel increasing the contact with the endothelium. Its compressible and secondary expandable properties make the foam suitable to fill tributary and collateral veins. Ethiodol facilitates the visualization under fluoroscopy by opacifying the mixture. 

#### 14.2.2. Glues and Non-Adhesive Liquid Embolics

In endovascular applications, besides sclerosants, the use of adhesive and non-adhesive liquid embolics has recently gained popularity. Cyanoacrylates are polymeric embolic agent with some advantages consisting of relatively precise control over placement and control of the rate of polymerization. Since the polymeric reaction is exothermic, giant cell reaction and fibrosis results likely contribute to its effectiveness. These features have been exploited in sclerotherapy of gastric varices, as well as in nonvascular systems, such as aneurysmal bone cysts and renal cysts. Contrarily, the limiting factor as a sclerosant is its embolic effect with a non-target action. Additionally, the use of cyanoacrylates requires the replacement of the delivery catheter after each administration, as it may occlude the catheter or cause its adhesion to native tissue [129]. The use of cyanoacrylates in embolotherapy are indicated for their inability to superselectively vessels cannulation, for long vessels segments embolization, or for patients in a hypocoagulable state. In particular, they are suited to cast vessels, with the advantages of a reduced risk of phlebitis, especially in the pampiniform plexus when used for varicocele sclero-embolization. Vascular malformations, acute hemorrhage, tumors, and venous disease are the most common applications of glues. Complications include tissue ischemia and systemic or local reactions. Cyanoacrylates are mixed with ethiodized oil to make them radiopaque and to adjust their polymerization time [131].

The variable polymerization time, due to specific properties of different commercially available compounds and the mixture with ethiodized oil, make cyanoacrylates generally less appreciated, compared to more recently developed non-adhesive embolics, such as ethylene vinyl alcohol (EVOH), although an advantage of glue is that it can be used in greater quantities, thanks to its low toxicity [5].

EVOH is a polymer dissolved in dimethyl sulfoxide (DMSO) that is increasingly used as a liquid embolic agent to treat a variety of conditions, including adult cerebral arteriovenous malformations (AVMs), dural arteriovenous fistulas (DAVFs), aneurysms, and hypervascular head and neck tumors. In July 2005, the Food and Drug Administration (FDA) approved the first commercial forms of EVOH with different concentrations (6% and 8%) in the United States for the presurgical embolization of brain AVMs in adults, but cautioned that its safety for use in children had not been studied, mainly due to DMSO toxicity. The suggested maximum injectable volume of Onyx is 0.545 mL/kg of DMSO [5]. Nevertheless, the initial success of Onyx in the adult population has resulted in its increased use in children, yet its safety and efficacy in the pediatric population have not been firmly established. Compared with traditional flow-directed catheters, the DMSO compatible catheters required for EVOH embolization are stiffer and less flexible at the tip and require a microwire for navigation, all factors that presumably increase the risk of vessel injury during catheterization. Furthermore, tortuous anatomy and small dysplastic feeding vessels can be difficult to navigate, especially in pediatric patients [42].

Among the non-adhesive liquid embolics, PHIL (precipitating hydrophobic injectable liquid) was the lastly introduced, featured by an iodine-bonded polymer dissolved in DMSO and free from metallic compounds. The advantages of PHIL, compared to Onyx, include that it does not require preparation and it is pre-loaded and ready to use; moreover, it produces very limited artifacts on imaging. A clinical trial has been recently registered to prospectively investigate the outcomes of intracranial dural arteriovenous fistulas treatment with PHIL in pediatric subjects (trial registration number NCT03731000) [132]. 

#### 14.2.3. Particles

The current intravascular particles include those composed of polyvinyl alcohol (PVA), which are made by a foam sheet vacuum dried and rasped into particles. The particles are filtered with sieves and are available in sizes ranging from 100 mm to 1100 mm. They are slightly irregular in size, facilitating the sorting through smaller sieve. PVA particles provide permanent occlusion by adherence to the vessel wall, causing stagnation of flow and focal angionecrosis [133]. Precisely calibrated particles emerged in the XXI century after FDA approval. Tris-acryl gelatin microspheres (TAGM) are available in six size ranges (40 to 120 µm, 100 to 300 µm, 300 to 500 µm, 500 to 700 µm, 700 to 900 µm, and 900 to 1200 µm) and packaged in prefilled syringes containing 2 mL of spheres in saline. They are made of an acrylic polymer matrix impregnated and embedded with porcine gelatin. They are non-resorbable hydrophilic particles; in addition, they can be temporarily compressed by 20 to 30% of their initial diameter. Unlike PVA particles, TAGM are smooth and spherical in shape, and fragmentation is not observed. In the last two decades, several manufacturers have developed their own calibrated particles with specific technologies, and some brands now even offer devices in unique size categories. The cornerstone in particles development was the engineering of device capable of taking-up and entrapping chemotherapy drugs and releasing them in a controlled way, even hours or weeks after transcatheter embolization, which has revolutionized the field of transarterial chemo- and radioembolization [133].

Major particle embolics limitations rely on the contrast volume used to deliver the particles and the risk of crossing a misdiagnosed arteriovenous shunt into the venous circulation, with subsequent paradoxical embolization through the heart [5]. It is important to mention that bleeding related to embolization was noted more easily for microspheres embolization; some hypotheses could be that particles spread in small venules through small fistulas may obstruct venous outflow; increased blood flow derived from residual feeders may lead to the rupture of patent collaterals; continuous particle injection under pressure into a wedged catheter may determine vessel rupture. In contrast with particles, the use of a liquid embolic material could be advantageous if arterial anatomy is favorable, with a rapid occlusion and the consequent benefit of saving time and radiation exposure. A single injection of glue or non-adhesive liquid embolics can also decrease the amount of contrast media needed for particle suspension, which is notable in the neonate patients. Particles are particularly appreciated for the preoperative embolization of hypervascular tumors. Early surgical tumor resection after preoperative embolization should be mandatory, in order to avoid the reconstitution of collaterals [88].

#### 14.2.4. Resorbable Agents

Gelatin foam (Gelfoam) is a biologic substance made from purified skin gelatin, usually available in sterile sheets and as a powder comprised of 40 to 60 mm particles. Sheets can be cut into a variety of shapes. Small pieces of 1 to 2 mm can be mixed with dilute contrast and injected as pledgets or prepared as slurry. Another standard use of gelatin foam is to form a small torpedo that can be injected into the target vessel for a more proximal occlusion. Gelatin foam particles, however, can aggregate or swell on hydration into larger particles. This substance causes mechanical obstruction, slowing blood flow and hastening thrombus formation. In addition, it provides a scaffold for clot formation. Gelatin embolization provides temporary vessel occlusion, allowing for recanalization in a few weeks. This can be either an advantage or disadvantage, depending on the clinical situation. Contrarily, it has been postulated that Gelfoam can be associated with infection, due to trapped air bubbles [133].

#### 14.2.5. Mechanical Devices

Mechanical devices, mainly represented by coils and plugs, provide proximal occlusion, decreasing blood flow, but not destroying the microcirculation, unless the vascular supply is terminal, as occurs in organs such as the brain, spleen, and kidneys or in pulmonary sequestration [5]. Since the advent of the Gianturco coil in 1975, transcatheter embolization of vascular structures has become a commonplace pediatric cardiovascular intervention. Coils have undergone significant evolution since their initial design and are now discriminated by a number of key features. They typically range in wire diameters from 0.010 to 0.052 inches, with varying lengths and varying coil or helical diameters. The original designs used highly antigenic natural wool fibers to promote thrombosis, which has been now replaced with non-antigenic synthetic fibers, such as Dacron. Coil deployment can be either free, as in the case of pushable devices, or controlled, with two different release modalities, mechanical or electrical. Controlled-released coils are preferred when there is a significant risk of coil displacement and non-target embolization, as in case of intracranial aneurysms.

Vascular plugs have features of both embolic materials and occlusive devices; they are braided devices of nitinol, intended for permanent occlusions. They have a relatively lower profile, compared to other occluders, and can be released in a controlled fashion. Traditional expandable nitinol mesh vascular occlusion devices are derived from the septal occluders commonly used in cardiology [133]. The main vascular plugs family includes several device designs that adapt to the clinical indication. First generation plugs were single-lobe, single-layer nitinol mesh in a cylindrical shape providing flow velocity reduction and coagulation induction. New generation devices have been designed as a multilayer nitinol mesh with a three-lobe design to increase the occlusive surface, induce faster vessel occlusion, and improve stability within the target vessel. Additionally, polytetrafluoroethylene (PTFE) layers have been inserted within the mesh by some manufacturers to increase the thrombotic effect [134]. All vascular plugs present radiopaque markers at the proximal and distal ends to help the precise delivery. For transcatheter embolization procedures, the devices can be delivered to the target arteries using guiding catheters, long sheaths, or through a 0.038-inch diagnostic catheter. For an adequate vessel wall apposition and to avoid migration of the device, it is recommended an oversizing by 30–50%. The available sizes range from 4 to 24 mm [134]. While coils can be deployed through very low-profile microcatheters, many conventional occlusion devices, such as plugs, require a relatively large delivery system or a stiff delivery cable that often precludes their use in small children. Catheters may kink while negotiating a tight curve, while stiffer catheters and delivery sheaths risk injuring abnormal vessels and could lead to potentially serious complications.

This instance has been partially overcome by the development of microvascular plugs (MVP) that can be deployed through microcatheters, although they are limited to medium-small size target vessels. Differently from conventional vascular plugs, the MVP is made of a nitinol framework partially covered by a polytetrafluoroethylene (PTFE) membrane at the proximal portion that promotes an immediate blood flow obstruction [12].

#### 14.2.6. Stents

Stent classification can be based on skeleton coverage, skeleton manufacturing, flexibility, expansion, drug containing, and biodegradability. Uncovered stents are used in stent-assisted, coil embolization strategy, which can be employed in wide neck saccular aneurysms, in order to prevent coil protrusion and migration. In this modality, the stent is usually implanted before coil placement. In selected cases with thin target arteries, coils may be placed before stent release. Retrievable stents placement, according to the waffle cone technique and developed for neuro-interventional procedures, have been used with the aim to avoid definitive stent implantation that would require long-term antiplatelet therapy. Exclusive endovascular vessel repair is possible with covered stents, which exclude the aneurysm, while preserving the flow. Covered stents require stable carrier systems and introducers to be placed at the origin of the target artery or hydrophilic guidewires to engage the efferent vessels and stiff guidewires to make easier the covered stent. Balloon-expandable covered stents are usually more rigid and less suitable for the tortuous visceral arteries than self-expandable grafts: in many cases, the artery is straightened to adapt to the stent losing its natural tortuous course with an increased risk of dissection and thrombosis. An extra-femoral approach and coronary stent grafts use can facilitate stent delivery. In particular, transaxillary or transomeral approaches may overcome the issue of an unfavorable origin angle of the target artery from the aorta [1]. Flow-diverting stents rely on the theoretical capacity to obtain aneurysm thrombosis, preserving patency of its efferent branches. Their peculiar design promotes a progressive thrombosis of the aneurysm by reducing the flow at the aneurysm neck, determining a turbulence which leads toward an increased blood viscosity within the sac with final aneurysm exclusion. A laminar flow into side branches, including those arising from the aneurysm, is preserved through the stent interstices, thanks to a neo-endothelization process. That is why they were initially developed for the treatment of cerebral aneurysms. They are characterized by higher flexibility and lower profile than peripheral covered stents. These features, in addition to higher porosity than bare metal stents, make them an ideal device for some VAAs with multiple side branches and with difficult anatomies [1].

#### 14.2.7. Resuscitative Endovascular Balloon Occlusion (REBOA)

The resuscitative endovascular balloon occlusion of the aorta (REBOA) device is constructed with a compliant balloon mounted to a catheter, which is then advanced through a short sheath without a guidewire. The concept of REBOA, which was first utilized by a surgeon in the Korean War [135,136], is that of a temporary percutaneous means of controlling massive truncal and pelvic hemorrhage. The most extensive pediatric REBOA study comes from Japan, on 19,467 pediatric patients under 18 years of age, 54 of which had a REBOA deployed with 15 of those patients under the age of 16. This younger patient cohort demonstrated an overall survival rate of 53.3%, compared to 38.5% of the adolescents aged 16–18 years old. Other clinical end points were not analyzed, nor were there complications such as embolisms, acute kidney injury (AKI), or other ischemic events. 

Due to the lack of clinical evidence, the REBOA device is not yet FDA-approved for pediatric patients. One issue with REBOA use in children is the smaller aortic diameter, in comparison to adults [137]. Several systematic reviews and meta-analyses have been conducted on REBOA catheters use, with the following findings: when inflated, REBOA results in an increase in cardiac afterload, proximal aortic blood pressure, and myocardial and cerebral perfusion; the complication rate was below 6%; animal studies revealed that ischemic complications occurred when inflation times exceed 30 min; no strong evidence exists for a statistically significant improvement of the mortality rate [138].

### 14.3. Radiation Protection

Historically, the Alliance for ‘‘Radiation Safety in Pediatric Imaging’’ launched, in 2008, the ‘‘Step Lightly’’ campaign for radiation safety in pediatric interventional radiology [139], and approximately ten years later, the ‘‘Have-A-Heart’’ campaign, with focus on radiation dose management of children with congenital and acquired heart diseases [140]. On these bases, a consensus emerged on caution in managing imaging examinations and image-guided procedures [141].

Removing X-ray detector grids allows us to reduce the radiation dose, mostly for children under 20 kg, with the double profit of noise reduction from scatter and spatial remodulation of fluoroscopy, due to prior geometrical magnification of images. So, images need to be acquired using a larger field of view. To keep the dose as low as reasonably achievable (ALARA), strict collimation is employed, in conjunction with minimum frame rate. Mathematic algorithms maintain a larger matrix of acquisition and satisfactory resolution [5]. Additionally, minimizing the distance between the patient and the image detector to avoid photons loss before reaching the detector and use of pulsed fluoroscopy with the lowest pulse rate constitute effective tools for radiation dose reduction. 

The risk of radiation-induced cancer is greater in children than adults, by a factor of 2–10. This is attributed to the fact that the pediatric population has a large number of rapidly diving cells and a longer life expectancy to express cancer risk. This is especially valid between the ages of 0 and 20 years.

Children are more radiosensitive than adults, regarding the development of specific cancer types, including leukemia, thyroid, skin, breast, and brain cancer.

Several epidemiological studies have demonstrated a link between cancer risk and exposure to ionizing radiation during childhood. Concerning fluoroscopy-guided interventional procedures, studies related to the estimation of cancer risk in pediatric patients who were subjected to cardiac catheterizations have reported an increased risk of breast cancer in females. In addition to the occurrence of stochastic effects, deterministic effects have also been observed in pediatric patients undergoing fluoroscopy-guided procedures, accounting for chronic dermatitis.

It is now well-known that exposure to ionizing radiation can cause complex damage at the cellular and molecular levels by inducing DNA lesions. Beels et al. [142] detected, in blood samples, a dose-dependent increase in gene damage after pediatric cardiac catheterizations. Moreover, chromosomal aberrations and micronucleus tests in peripheral blood lymphocytes revealed chromosomal damage in pediatric patients with CHDs who underwent a series of cardiac imaging examinations [141].

## 15. Conclusions

Even though the majority of materials are shared with adult interventions, the smaller size and the greater delicacy of the body structures, in particular, of the vascular system, require special care and the predilection of low-profile devices.

## Figures and Tables

**Figure 1 jcm-11-06626-f001:**
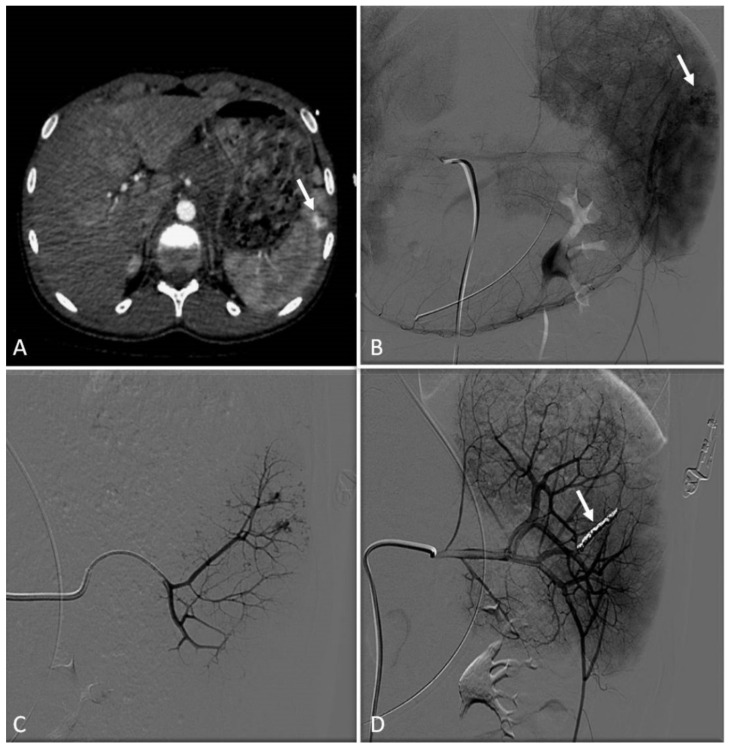
Transcatheter coil embolization of post-traumatic splenic bleeding in a 13-year-old female. (**A**) Axial CTA image shows a small pseudoaneurysm (arrow) along the lateral spleen margin. (**B**) DSA performed through selective 4F catheterization of the celiac trunk shows several small pseudoaneurysms within the spleen parenchyma (arrows). (**C**) DSA image of superselective coaxial 1.9F microcatheterization of the affected spleen vessel. (**D**) Control DSA image after superselective multiple 4-mm coils (arrow) embolization shows disappearance of the pseudoaneurysms, with preserved opacification of the surrounding spleen parenchyma. No further bleeding occurred, nor were ischemic complications observed.

**Figure 2 jcm-11-06626-f002:**
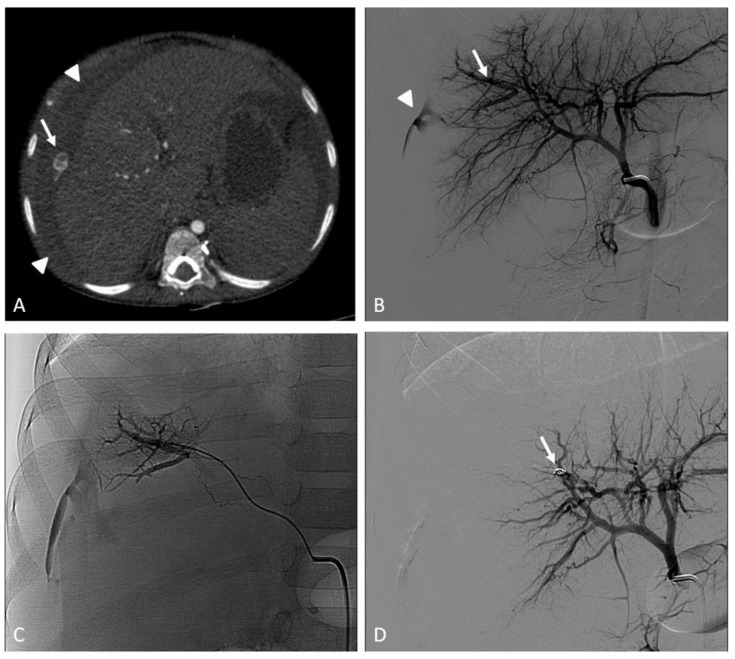
Transcatheter particle embolization of iatrogenic hepatic bleeding after percutaneous needle biopsy in a 9-year-old male. (**A**) Axial CTA image shows a perihepatic fluid collection (arrowheads) with contrast medium extravasation (arrow), consistent with active arterial bleeding from the site of percutaneous biopsy in the right liver lobe. (**B**) Selective hepatic artery digital subtraction angiography (DSA) performed with a 4F catheter shows the presence of an arterio-portal shunt with opacification of peripheral right portal branches (arrow) and contrast extravasation (arrowhead) from the lateral hepatic margin. (**C**) Fluoroscopic image shows superselective coaxial 1.9F microcatheterization of the artery feeding the arterio-portal shunt and the bleeding. (**D**) Selective hepatic artery DSA demonstrating disappearance of the bleeding after 3-mm coil (arrow) embolization.

**Figure 3 jcm-11-06626-f003:**
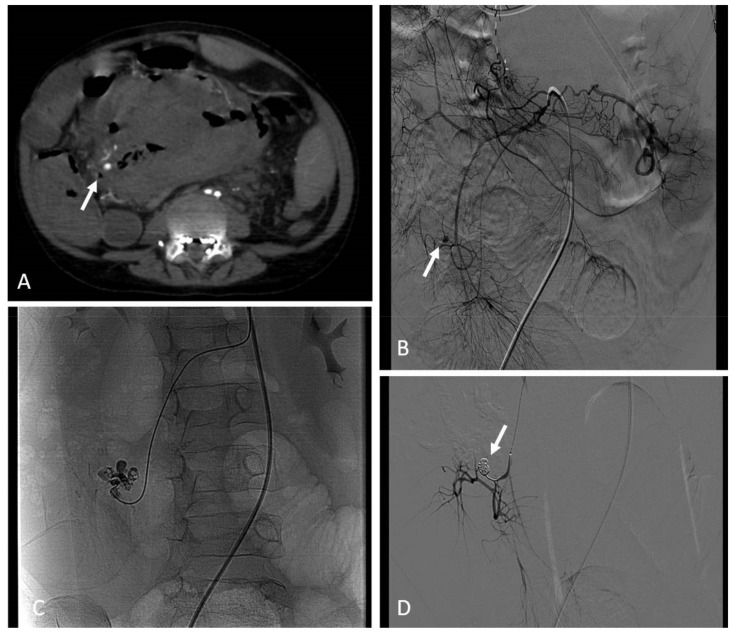
Transcatheter coil embolization of spontaneous abdominal bleeding in a lymphoproliferative disorder of the ileum in a 10-year-old female. (**A**) Axial CTA image shows a large necrotic mass with hyperdense component, consistent with recent bleeding within the mesentery, with a small arterial pseudoaneurysm (arrow). (**B**) Selective superior mesenteric artery DSA performed with a 4F catheter shows the small pseudoaneurysm (arrow) affecting the ileocolic artery. (**C**) Fluoroscopic image shows contrast extravasation due to pseudoaneurysm rupture during superselective coaxial 2.7F microcatheterization of the ileocolic artery. (**D**) Control DSA image after superselective 3-mm coil (arrow) embolization shows disappearance of the pseudoaneurysm with no longer contrast extravasation.

**Figure 4 jcm-11-06626-f004:**
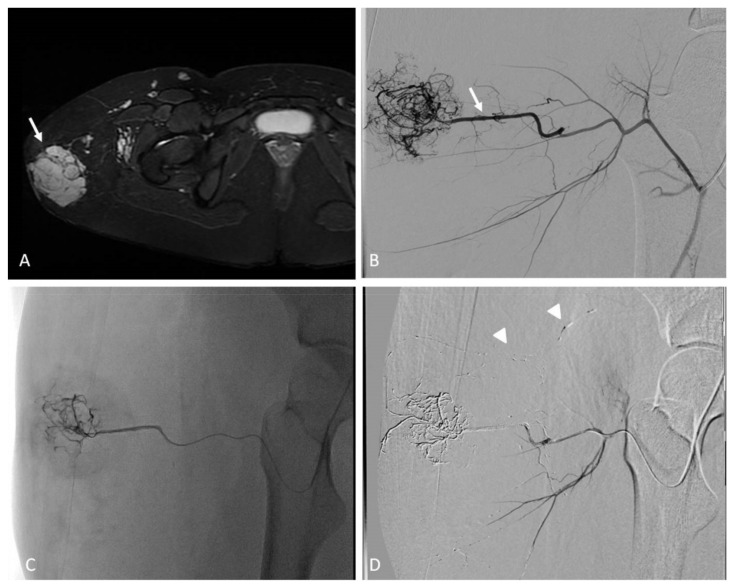
Transcatheter ethylene vinyl alcohol (EVOH) copolymer embolization of a simple high-flow vascular malformation of the thigh in a 16-year-old female. (**A**) Magnetic Resonance Angiography T2-weighted image shows a hyperintense mass consistent with a vascular malformation (arrow). (**B**) Selective DSA of the deep femoral artery confirms the high-flow nature of the vascular malformation with a prevalent feeder (arrow). (**C**) Fluoroscopy image, acquired during EVOH embolization after superselective 2.7F coaxial microcatheterization of the lateral circumflex artery, shows capillary distribution of the embolic agent throughout the lesion. (**D**) Control DSA confirms the complete devascularization of the malformation; note little embolic material spread into the draining veins of the lesion (arrowheads).

**Figure 5 jcm-11-06626-f005:**
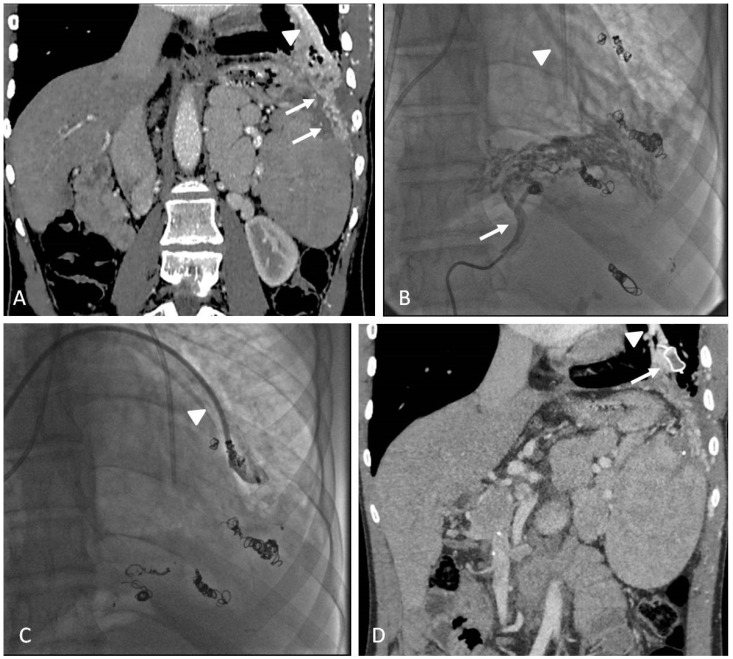
Combined transarterial and transvenous transcatheter coil and plug embolization of a complex iatrogenic high-flow vascular malformation with systemic to pulmonary shunt in a 17-year-old male. A large left diaphragmatic high-flow vascular malformation developed after iatrogenic splenic infarction and abscess. (**A**) Reformatted coronal Computed Tomography Angiography (CTA) image shows the vascular malformation (arrows) fed by several arteries, among which intercostal, lumbar, diaphragmatic, and left gastric, draining into the left inferior pulmonary vein (arrowhead). The presence of a direct shunt with the pulmonary vein did not allow us to inject liquid embolics and particles in the afferent arteries, due to the risk of non-target cerebrovascular and systemic embolization. (**B**,**C**) Fluoroscopic images show the complete evaluation of the inflow and outflow of the malformation, realized through direct transarterial catheterization (arrow) and transeptal US-guided puncture of the heart to select the left inferior pulmonary vein (arrowheads); embolization of the arteries was performed with several metallic coils and plugs. (**D**) Reformatted coronal CTA image shows the plug (arrow) in the left inferior pulmonary vein (arrowhead) with reduced shunt.

**Figure 6 jcm-11-06626-f006:**
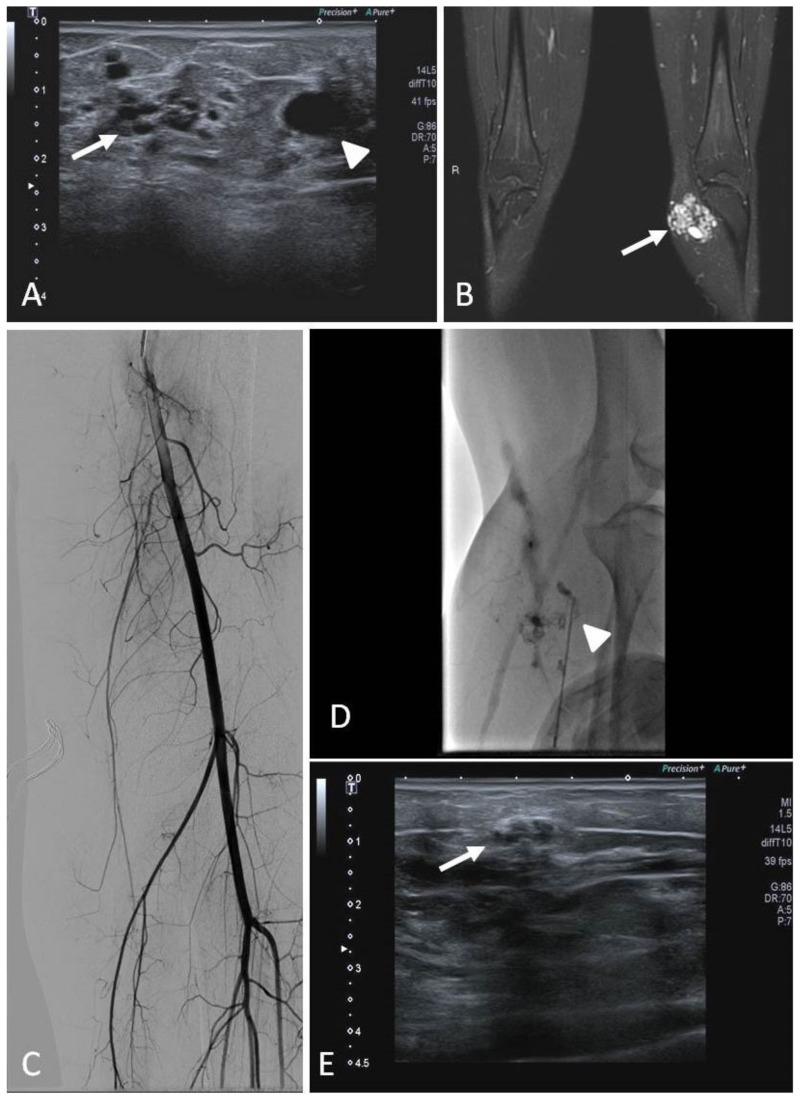
Percutaneous sclero-embolization of a simple low-flow vascular malformation of the leg in a 13-year-old female. (**A**) B-mode Ultrasound (US) image shows a tangle of subcutaneous varicose vessels (arrow) draining into a dilated great saphenous vein (arrowhead). (**B**) Coronal T2-weighted Magnetic Resonance image shows an hyperintense mass (arrow) consistent with a vascular malformation. (**C**) DSA of the femoral axis demonstrates the low-flow nature of the lesion that has no significant arterial feeders. (**D**) Fluoroscopic image acquired during US-guided fine needle (arrowhead) puncture of the dilated vessels shows the distribution of the injected mixture of detergent sclerosant and ethiodized oil (to convey radiopacity); a tourniquet was tied on the thigh (not shown) to stop the saphenous flow and prevent pulmonary embolism. (**E**) B-mode US image shows post-procedural hyperechoic appearance with acoustic shadow of the sclerosed veins (arrow).

**Figure 7 jcm-11-06626-f007:**
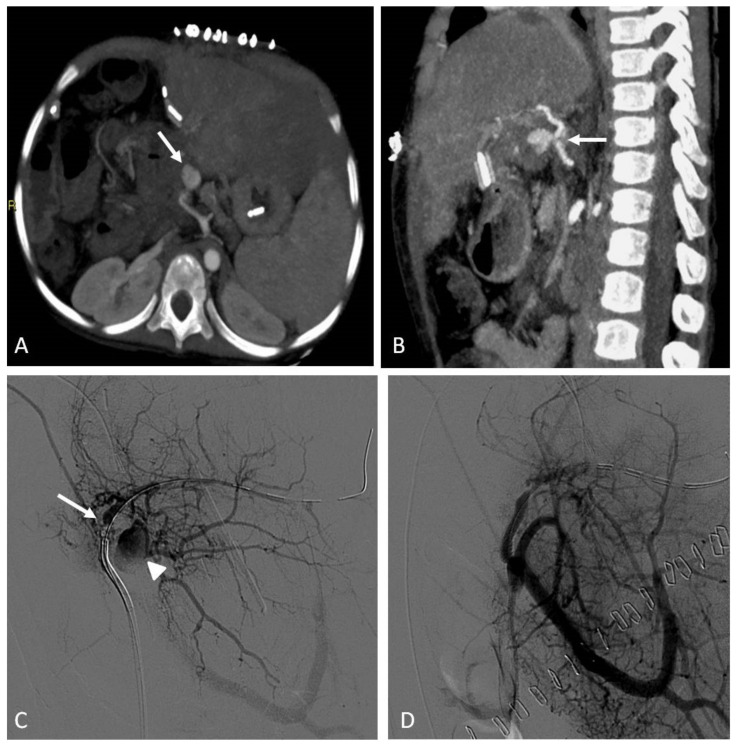
Transcatheter repair with coronary stent graft of a hepatic artery pseudoaneurysm after liver transplant and meso-rex surgery in a 6-year-old female. (**A**) Axial CTA image shows a pseudoaneurysm of the proximal tract of the transplanted hepatic artery (arrow). (**B**) Reformatted coronal CTA image highlights the presence of a hepatic artery stenosis (arrow), associated with the pseudoaneurysm. (**C**) DSA image shows selective 5F catheterization of the hepatic artery, confirming the presence of stenosis (arrow) and pseudoaneurysm (arrowhead); a 0.014” guidewire was advanced beyond the affected tract. (**D**) Control DSA, performed after deployment of two imbricated coronary 3.5-mm balloon expandable stent grafts, shows regular opacification of the artery and disappearance of the pseudoaneurysm.

**Figure 8 jcm-11-06626-f008:**
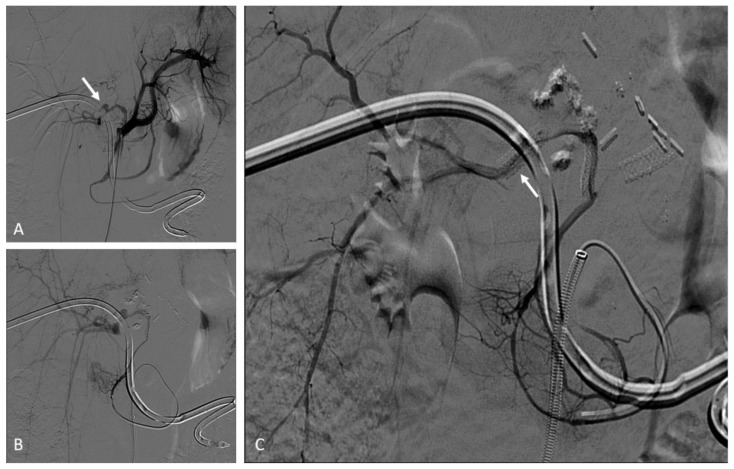
Transcatheter repair with coronary stent graft of a hepatic artery pseudoaneurysm after liver transplant and percutaneous biliary drainage in a 9-year-old female presenting with hemobilia. (**A**) DSA image shows a small pseudoaneurysm (arrow) of the hepatic artery close to the percutaneous biliary drainage catheter. The common hepatic artery is thrombosed, and the proper hepatic artery is revascularized through the gastroduodenal artery, due to a previous endovascular procedure for anastomotic stenosis. (**B**) DSA image shows superselective coaxial 1.9F microcatheterization of the gastroduodenal artery through a 5F introducer sheath and a 5F coaxial guiding catheter placed at the origin of the superior mesenteric artery. (**C**) Control DSA, performed after deployment of a coronary 3-mm balloon expandable stent graft (arrow), shows regular opacification of the artery and disappearance of the pseudoaneurysm.

**Figure 9 jcm-11-06626-f009:**
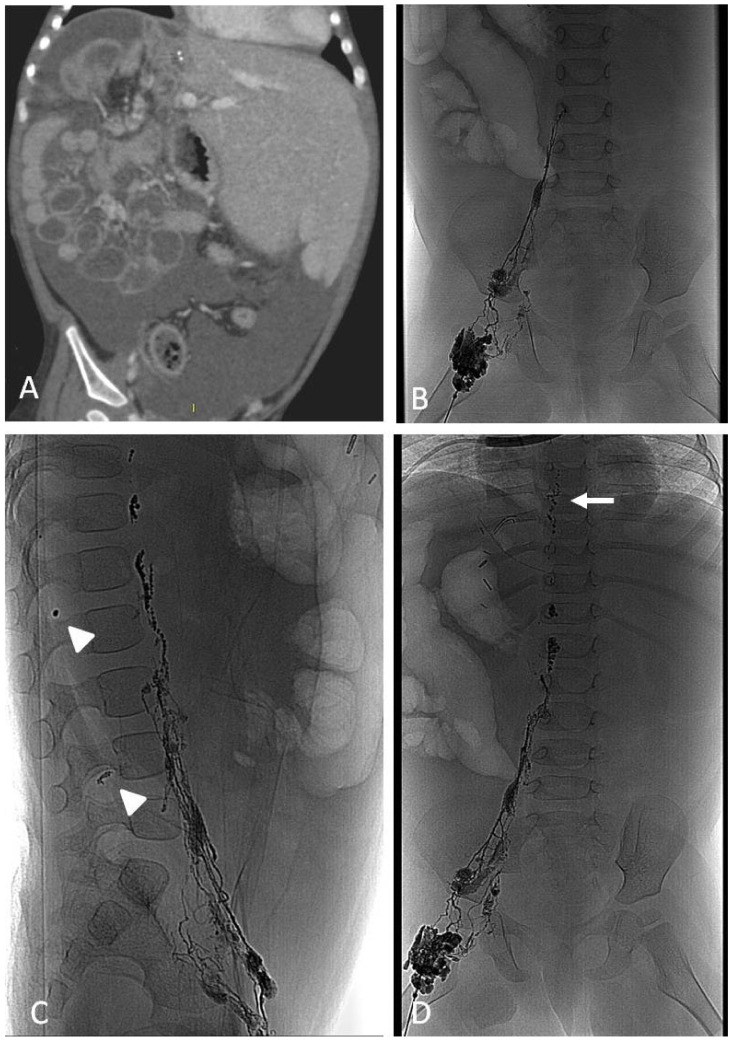
Percutaneous lymphangiography and sclero-embolization of chilous ascites, due to lymphatic leakage after liver transplant in a 3-year-old female. (**A**) Reformatted coronal Computed Tomography image shows abundant ascites. (**B**) Fluoroscopic image shows the ethiodized oil contrast medium injected into the groin lymph nodes spreading through the iliac lymphatics. (**C**) Abdominal X-rays lateral view shows extravasation (arrowheads) of ethiodized oil confirming the lymphatic leakage. (**D**) X-rays frontal panoramic view shows partial opacification of the lymphatic ducts up to the thoracic duct (arrow). After percutaneous lymphangiography the ascites progressively reduced up to complete resolution in few weeks.

**Figure 10 jcm-11-06626-f010:**
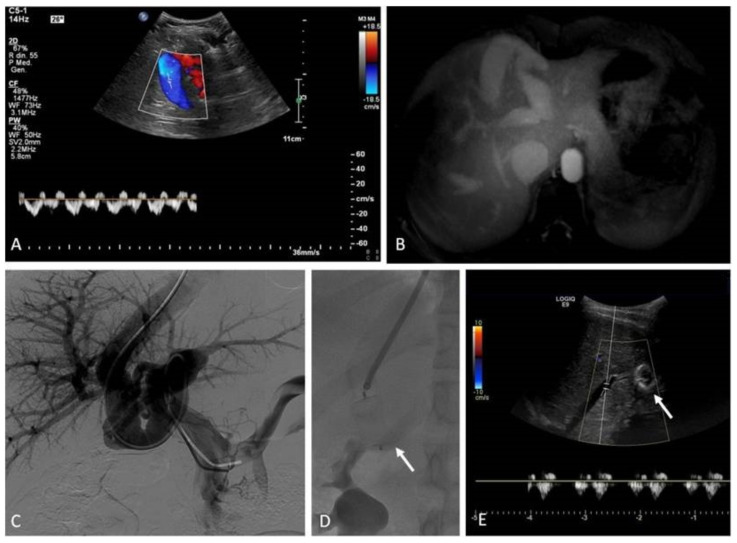
Transcatheter plug occlusion of a congenital intrahepatic porto-systemic shunt in a 17-year-old female. (**A**) CDUS image shows an enlarged middle hepatic vein aberrantly communicating with the left portal vein branch. (**B**) Magnetic Resonance Angiography image offers a panoramic view of the aberrant porto-systemic shunt, useful to plan the percutaneous occlusion procedure. (**C**) Transjugular portal vein venography was performed through retrograde catheterization of the shunt. (**D**) Fluoroscopic image shows the deployment of a 24-mm vascular plug (arrow) within the shunt. (**E**) B-mode US image shows size normalization of the middle hepatic vein after effective shunt occlusion with the plug (arrow). The patient healed from post-prandial hyperammonemia.

**Figure 11 jcm-11-06626-f011:**
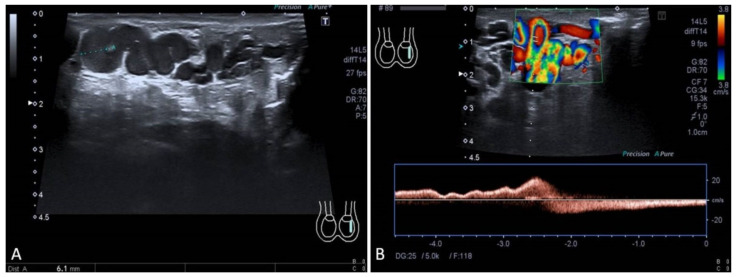
Color Doppler Ultrasound findings of recurrent varicocele after surgery in a 16-year-old male. (**A**) B-mode US image shows high grade ectasia of the left pampiniform plexus extended to the inferior pole of testicle. (**B**) Color Doppler US examination shows persistent flow reversal during the Valsalva maneuver.

**Figure 12 jcm-11-06626-f012:**
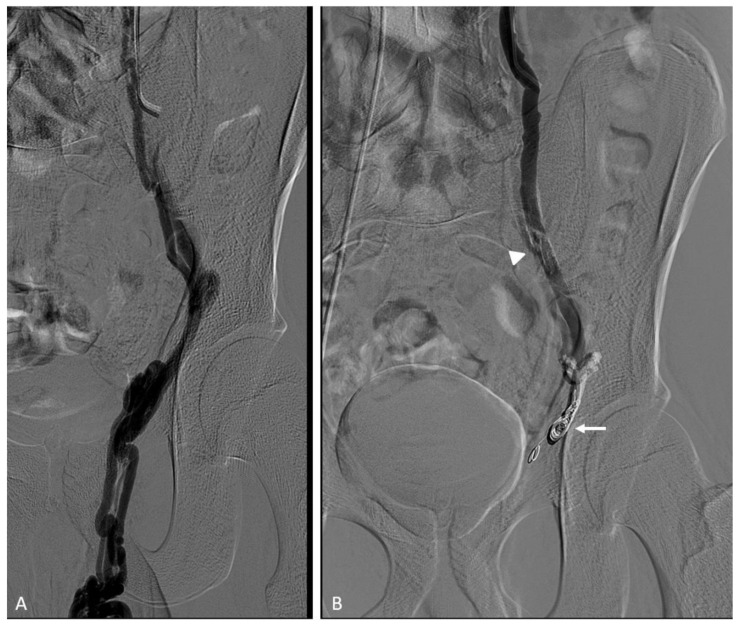
Transcatheter coil and glue embolization of recurrent varicocele after surgery in a 16-year-old male. (**A**) Selective left spermatic vein venography image shows an enlarged and refluent left spermatic vein with duplication and retrograde opacification of the pampiniform plexus. (**B**) Post-embolization venography image proves absence of opacification of the pampiniform plexus after distal occlusion with coils (arrow) and glue that spread into a duplicated vein (arrowhead).

**Figure 13 jcm-11-06626-f013:**
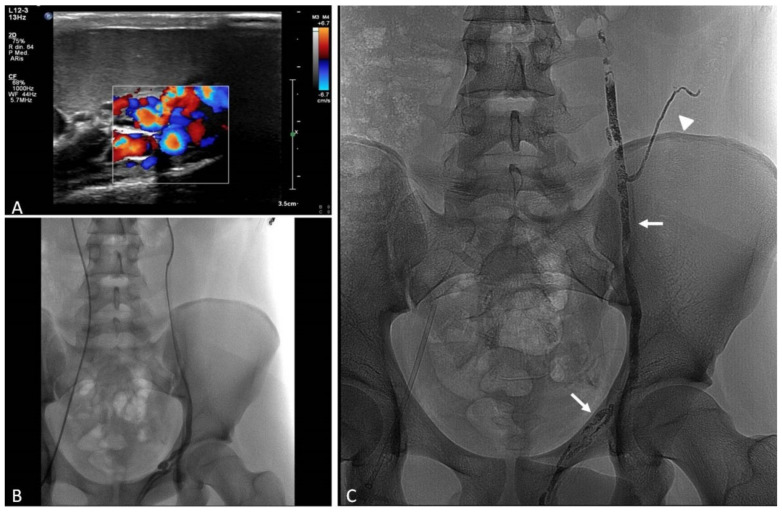
Transcatheter glue embolization of varicocele in a 17-year-old male. (**A**) Color Doppler US image shows enlarged vessels of the pampiniform plexus with flow reversal during the Valsalva maneuver. (**B**) Fluoroscopic image shows selective transfemoral catheterization of the left spermatic vein up to the iliac region above the inguinal canal. (**C**) X-rays image shows the glue/ethiodized oil cast completely filling the left spermatic vein. Note the duplicated spermatic veins (arrows) and the collateral vessel (arrowhead) also filled with the embolic material.

## Data Availability

No new data were created or analyzed in this study.

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
