# Peer review of "Embolization in Pediatric Patients: A Comprehensive Review of Indications, Procedures, and Clinical Outcomes"

_jcm, 2022, doi:10.3390/jcm11226626_

Round 1

Reviewer 1 Report

The authors presented a narrative review regarding the indication, techniques and corresponding clinical outcomes of the embolization procedures in pediatric patients. This is an very nicely written review that describes a broad range of diseases. The review gives a global overview of the various strategies that can be applied in a broad range of diseases, although this a disadvantage in itself.  Multiple of the topics are in itself complex enough to warrant its own review of the various techniques and outcomes. It is therefore more of a global overview and lacks the granularity to guide treatments. The aims of the review are formulated and the key statements are supported by references. The presentation and selection of data are appropriate.

Comments:

1.     The various sections do not have the same structure and misses consistency within the various sections. For instance, techniques under cardiovascular system and pulmonary vascular malformations is discussed under ‘Indications’.

2.     Clinical outcomes are furthermore missing in various sections. Embolization of for instance intra-axial tumours is for instance technically feasibility, but generally a bad idea and without evidence.

Minor comments:

  1. Mention the level of evidence in each sub-area of the disease if possible. 

  1. The repetitive use of ‘Embolization in’ within the section titles is unnecessary. The title and introduction of the paper already shows that it’s regarding embolization.

  1. The section cardiovascular system and pulmonary vascular malformations could be separated in two to enhance clarity.

  1. Please provide the literature search in detail if possible, including the search terms and inclusion criteria with Methods part.

Reviewer 2 Report

Authors have extensively reviewed endovascular procedures in pediatric patients. Although the article is long and lack in descriptions about some rare entities such as intracranial AV shunts peculiar to pediatric patients, I commend the authors for their large variety of clinical experiences.

For readability, I would recommend describe general considerations in pediatric patients first and then further mention each disease.

Description about radiation dose is slightly limited. Possible adverse effects in the long-term from increased radiation dose might be added.
